# Task-agnostic Prompt Compression with Context-aware Sentence Embedding and Reward-guided Task Descriptor

**Barys Liskavets**[*1], **Shuvendu Roy**[2], **Maxim Ushakov**[1],
**Mark Klibanov**[3], **Ali Etemad**[2], **Shane K. Luke**[3]
[1]**Alterra AI, Palo Alto, United States,**  [2]**Queen's University, Canada,**  [3]**Workday Inc**

**Reviewed on OpenReview:** `https://openreview.net/forum?id=TpGcX9UTOt`

## Abstract

The rise of Large Language Models (LLMs) has led to significant interest in prompt compression, a technique aimed at reducing the length of input prompts while preserving critical information. However, the prominent approaches in prompt compression often require explicit questions or handcrafted templates for compression, limiting their generalizability. We propose Task-agnostic Prompt Compression (TPC), a novel framework that generalizes compression across tasks and domains without requiring input questions or templates. TPC generates a context-relevant task description using a task descriptor trained on a curated dataset of context and query pairs, and fine-tuned via reinforcement learning with a reward function designed to capture the most relevant information. The task descriptor is then utilized to compute the relevance of each sentence in the prompt to generate the compressed prompt. We introduce 3 model sizes (Base, Large, and Huge), where the largest model outperforms the existing state-of-the-art methods on LongBench and ZeroSCROLLS, and our smallest model performs comparable to the existing solutions while being considerably smaller. Finally, we release the code and the dataset for quick reproducibility and further development: `https://github.com/bliskavets/TPC`.

## 1 Introduction

The emergence of Large Language Models (LLMs) has spurred extensive research into prompting techniques, including chain-of-thought reasoning Wei et al. (2022), in-context learning Dong et al. (2024), and retrieval-augmented generation Lewis et al. (2020) to harness LLMs' generalization and reasoning abilities for various applications. In practice, effective prompting often requires detailed and lengthy inputs to generate high-quality responses in domain-specific tasks. However, long prompts significantly increase inference time and costs. To address this, a new research direction called *prompt compression* Jiang et al. (2023b;c); Liskavets et al. (2025) has emerged, which focuses on reducing prompt length while retaining the critical information needed to accurately answer user queries.

Early research on prompt compression predominantly focused on token-level techniques, which involve analyzing the importance of individual tokens in a prompt and removing less informative ones to generate the compressed prompt. However, token-level compression often leads to incoherent or fragmented sentences, hindering the overall performance. To address this, recent approaches, such as CPC Liskavets et al. (2025), have shifted to sentence-level compression, which evaluates the relevance of each sentence to the input question. While this method represents a significant improvement over token-level approaches, its reliance on explicit questions or manually crafted templates for non-question-based tasks (e.g. summarization or code) restricts its broader applicability.

In this work, we propose Task-agnostic Prompt Compression (TPC), a generic prompt compression method capable of compressing input prompts across tasks and domains without relying on input questions or

---

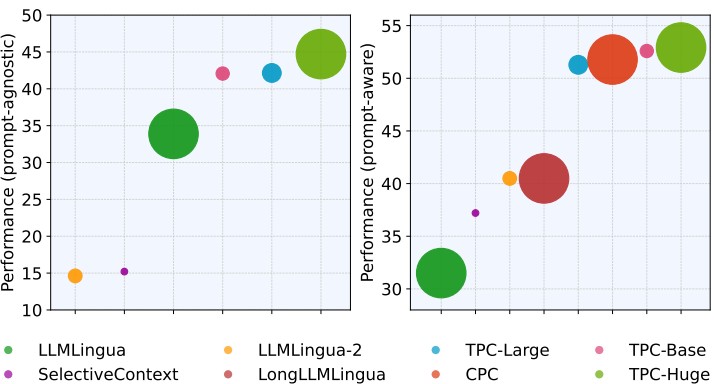

Figure 1: Comparison of model size versus performance for different prompt compression methods in both task-aware and task-agnostic setups. Our largest model, TPC-Huge, outperforms all existing methods while maintaining a comparable size to existing solutions. On the other hand, our smallest model, TPC-Base, achieves a competitive performance despite being significantly smaller in size.

handcrafted prompt templates. The central idea of TPC involves using a novel task descriptor to create a context-relevant task description, which facilitates compression by comparing its context-aware embedding similarity to each sentence in the prompt. Here, the task descriptor is a causal LM, trained on our curated dataset of prompt-question pairs, that generates a context-relevant task description encapsulating the main concept of the prompt. Since the effectiveness of the prompt compression relies on the quality of the generated task description, we introduce a reinforcement learning (RL) approach to further fine-tune the task descriptor with a novel reward function designed to encourage capturing the most relevant information of the prompt in the task description. Finally, the context-aware embedding is computed using a context-aware sentence encoder trained on our curated dataset of multi-hop questions, answers, and positive and negative pairs.

We present comprehensive experiments to evaluate our proposed solution and compare it against existing methods on two evaluation setups: task-aware compression and task-agnostic compression on two popular benchmarks LongBench and ZeroSCROLLS adopted by existing literature on prompt compression. We report the results for three variants of our model size: TPC-Base, TPC-Large, and TPC-Huge, containing 0.5B, 1B, and 7B parameters, respectively. As summarized in Figure 1, our TPC shows significant improvements over existing methods on both task-aware and task-agnostic setups. Our smallest model, while being considerably smaller in size, outperforms or performs comparable to the existing state-of-the-art (SOTA) methods. Overall, we make the following contribution in this paper:

- We introduce TPC, a task-invariant prompt compression method, which unlike existing methods doesn't require task-specific hand-crafted templates to achieve generalizable compression. TPC generates a context-relevant task description using a task descriptor LM, which is then used for compression by comparing the context-relevant embedding to each sentence in the prompt.
- We develop a synergistic training framework consisting of two components: a contextual task description (CTD) module which generates concise task-relevant descriptions that guide the contextual segment importance scoring module (CSE). Both modules share the same language-model backbone with only a lightweight LoRA head added. Furthermore, we introduce a novel compression-quality reward combined with RL-based fine-tuning to improve CTD quality, resulting in more robust compression and stronger out-of-distribution generalization.
- We curate two datasets required for training the context-relevant task descriptor and context-aware sentence encoder of our proposed TPC.
- We propose *task-agnostic prompt compression* as a new problem formulation, where the compression method must work universally without manual user intervention via hand-crafted prompting. Our proposed solution matches or outperforms existing methods on both task-aware and task-agnostic compression setups, while using up to 14× fewer parameters (0.5B vs. 7B) and shows strong generalization across tasks and domains. To enable quick reproduction and further development, we will release the datasets and the codebase upon acceptance.

## 2 Related Works

### 2.1 Prompt Compression

Recent efforts in prompt compression focus on reducing the inference cost of LLMs. A key line of research involves model-agnostic methods, which leverage pre-trained models for compression. Early works like token pruning during the forward pass (Kim et al., 2022) and recursive context summarization (Chen et al., 2023) introduced effective strategies but required access to the pre-trained LLM, which is often impractical. To address this limitation, newer approaches such as LLMLingua (Jiang et al., 2023b) utilize token-level perplexities to filter out semantically insignificant tokens, while Selective-Context (Li et al., 2023a) retains high-perplexity tokens using decoder-only models like LLaMa and GPT-2. LongLLMLingua (Jiang et al., 2023c) further refines this idea by integrating question relevance for context-aware compression. However, these methods often lack adaptability to new domains, restricting their broader applicability.

In parallel, trainable methods for prompt compression have also gained traction as a key research direction. Soft prompt techniques (Wang et al., 2024; Bulatov et al., 2022; Chevalier et al., 2023) fine-tune or pre-train LLMs to achieve high compression rates, though they provide limited interpretability and control over the compression ratio. Sequence-to-sequence models compress context by generating a summary directly (Xu et al., 2024), but their autoregressive design introduces latency. Reinforcement learning approaches, such as optimizing for simplicity, fluency, and salience (Laban et al., 2021), or using compression ratio as a reward (Jung & Kim, 2024), offer an alternative, but may fall short in task-aware tasks. Recent innovations, such as (Pan et al., 2024), propose models that evaluate and prune tokens based on their information value, providing a more systematic approach to compression. Despite these advances, challenges remain in balancing efficiency, accuracy, and domain adaptability. More recently CPC Liskavets et al. (2025) proposed to utilize a context-aware sentence encoder to find the relevance of each sentence in the context to remove irrelevant sentences from the input prompt, achieving SOTA performance on existing benchmarks. However, a major limitation of such an approach is its reliance on the input question to guide the compression, requiring manual prompt formats for different tasks. While there have been some efforts towards building task-agnostic prompt compression Pan et al. (2024), the performance of such methods falls short of the task-aware counterpart. Our goal in this work is to develop a task-agnostic prompt compression method without sacrificing performance.

### 2.2 Sentence Embedding Learning

Sentence embedding learning aims to create high-dimensional vector representations of text that capture semantic meaning, facilitating tasks such as text classification, clustering, and similarity search. Early approaches like GloVe (Pennington et al., 2014) and Word2Vec (Mikolov et al., 2013) focused on word- or token-level embeddings. Later, sentence-level representation learning gained attention, with works like Reimers & Gurevych (2019) fine-tuning BERT-based models (Vaswani et al., 2017) to extract sentence embeddings for measuring text similarities. Subsequent research (Li et al., 2023b; Wang et al., 2022; Beltagy et al., 2020) enhanced the effectiveness of these embeddings, particularly for longer contexts. Later, BehnamGhader et al. (2024) leveraged the extensive knowledge of pre-trained LLMs to build robust sentence encoders. However, while these models excel at sentence representation, they lack context awareness, which is an essential property for our prompt compression approach. While a more recent work, CPC (Liskavets et al., 2025) explored context-aware sentence encoding to compress prompts based on their relevance to the question, the proposed encoder was not task-agnostic. In this work, we introduce a task-agnostic and context-aware sentence encoder, where the embedding captures the semantics of the context.

## 3 Proposed Method

### 3.1 Problem Statement

The goal of prompt compression is to transform a long input prompt into a shorter version while preserving all the information necessary for the LLM to produce an answer of comparable quality to that generated

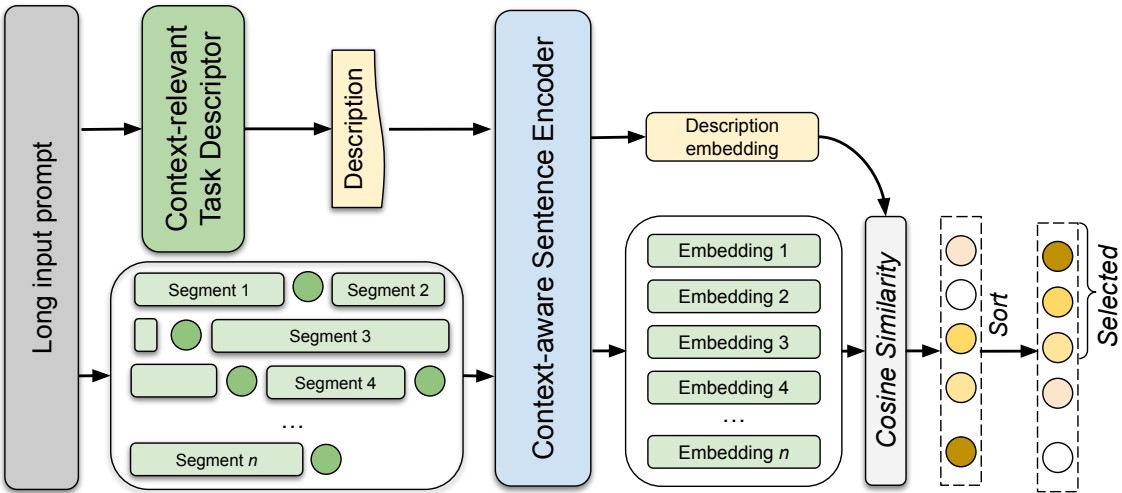

Figure 2: Illustration of our proposed prompt compression method. The CTD module generates a task description that is relevant to the input context. This description is then utilized by the Context-Aware Sentence Encoder to evaluate the relevance of each sentence in the input prompt, ultimately generating the compressed prompt.

from the original prompt. Most prior works on prompt compression expect both the text and an explicit question as input, using the question to assess the relevance of different text segments. This becomes a limitation, since in many real-world scenarios, users of prompt compression methods may not be able or may not wish to formulate an appropriate question to accompany the text. In contrast, our method enables context compression without additional manual input or task-specific configurations. In our work, we define a task-agnostic setup, where the prompt compression method receives a text input without any explicitly marked components that can or cannot be compressed. The compression model must automatically determine which parts of the prompt are important and to what extent they should be compressed. In this setup, the model must infer the domain and structure of the prompt autonomously.

### 3.2   Task-agnostic Prompt Compression

The conventional setup of task-aware prompt compression operates by leveraging an explicit question (or a hand-crafted prompt where no question is available, e.g. code) $q$ provided along with the input context $c$. Specifically, this approach utilizes a sentence encoder $f_s$ to encode both the question and individual sentences $\{s_1, .., s_n\}$ from the context into embeddings $\mathbf{e}_q = f_s(q)$ and $\mathbf{e}_{s_1}..\mathbf{e}_{s_n} = f_s(s_1, ..., s_n)$, respectively. The relevance of each sentence is then determined by a scoring function (e.g. cosine similarity) $\mathcal{R}(\mathbf{e}_q, \mathbf{e}_{s_i})$, to select the top $k$ most relevant sentences:

$$\mathcal{S} = \phi(q, c) = \text{TopK}(\{\mathcal{R}(\mathbf{e}_q, \mathbf{e}_{s_i}) \mid s_i \in c\}), \tag{1}$$

where $\mathcal{S}$ is the compressed prompt containing the selected sentences. The encoder $f_s$ is trained on a dataset of $(q, c, pos, \{neg^{i=1..m}\})$ tuples to learn context-aware sentence embeddings, where $q$ is a question, $c$ is the context, and $pos$ is the positive sentence, and $neg^{i=1..m}$ are negative sentences. Here, a positive is a sentence containing relevant information to the question, while the negatives are context sentences containing no information regarding the question. While effective, this approach inherently relies on the availability of explicit questions and cannot be adapted to tasks with no explicit questions, such as summarization.

To address the limitations of task-aware compression, we propose Task-agnostic Prompt Compression (TPC), a novel solution for prompt compression that does not require such explicit questions. Instead, our approach generates a context-relevant task-description $\hat{q}$ from the input prompt $p$ using our proposed Context-relevant Task Descriptor (CTD) model $f_q$ as $\hat{q} = f_q(p; \theta_q)$. CTD is a lightweight module designed to generate a task description that highlights the most relevant information within $p$. The task description $\hat{q}$ is then used to find

the relevance of each sentence in the input prompt with our context-aware sentence encoder. To ensure the generated task description is semantically meaningful and contextually relevant, we train $f_q$ on a synthetic dataset of $(p, q)$ pairs (curated using our proposed data curation pipeline discussed in Section 3.3), followed by a reward-guided fine-tuning stage with our proposed reward function (Section 3.4). Finally, we propose a new method for training the Context-aware Sentence Encoder (CSE) that effectively captures the context in the embedding (Section 3.5). Note that the CTD in our proposed method is a small LLM to generate the task description only; the final response is generated by a pre-trained LLM (like the existing method). Overall, the lightweight CTD add negligible overhead compared to the computation cost of the LLM. The overall diagram of our proposed method is illustrated in Figure 2, and a pseudo-code is provided in Algorithm 3 (Appendix 3.7).

## 3.3   Context-relevant Task Descriptor

CTD is the key component of our proposed solution that transforms a task-agnostic prompt compression task into a task-aware prompt compression task. Specifically, CTD generates a context-aware task description $\hat{q}$ from the input prompt $p$. The CTD comprises two primary components: (1) a dataset curation pipeline for generating high-quality $(p, q)$ pairs, and (2) a causal encoder training process on the curated dataset to generate the task description $\hat{q}$. Below, we describe these components in detail.

### 3.3.1   Dataset Curation

To train the CTD, we require a dataset of $(p, q)$ pairs, where $p$ represents a long input prompt and $q$ is a relevant query or description that highlights the essential information in $p$. We leverage an existing dataset of texts $D_{init}$ and a pre-trained LLM to generate these pairs. Specifically, for a given input prompt $p$, the LLM is prompted to generate $q$ using the pre-designed Prompt 1 in A.2. By applying this prompt to a dataset of long texts (e.g. Wikipedia), we generate a collection of question and input context pairs.

In real-world scenarios, user prompts are often structured, typically comprising an input context, a question, and a system prompt. To ensure that our generated dataset adheres to such a structure and remains suitable for downstream tasks, we refine the initially generated dataset by prompting the same pre-trained LLM with Prompt 2 in A.2. This two-stage process yields a dataset $\mathcal{D}_{\text{CTD}}$ of input prompts and structured task descriptions, which is subsequently used to train the question generator $f_q$, as described in the next subsection. Additionally, we provide a few examples from the resulting training dataset in the Appendix.

### 3.3.2   Training CTD

At the core of CTD is a causal encoder, $f_q$, designed to generate contextually relevant task descriptions. We utilize a causal language model for $f_q$, initialized from a pre-trained LLM, to auto-regressively generate the task description $\hat{q}$ conditioned on the input prompt $p$. Formally, given input long prompt $p$, $f_q$ generates the question $\hat{q}$ as follows:

$$P(q \mid p; \theta_q) = \prod_{t=1}^{T} P(q_t \mid q_{<t}, p; \theta_q), \tag{2}$$

where $q_t$ is the $t$-th token in $q$, $q_{<t}$ represents the sequence of tokens generated up to step $t - 1$, and $\theta_q$ are the model parameters. We train the encoder $f_q$ to maximize the likelihood of generating the target question $q$ given the prompt $p$. The Supervised Training (ST) procedure is carried out using the following loss function:

$$\mathcal{L} = -\mathbb{E}_{(p,q) \sim \mathcal{D}'_{\text{CTD}}} \left[ \sum_{t=1}^{T} \log P(q_t \mid q_{<t}, p; \theta_q) \right], \tag{3}$$

where $T$ is the total number of tokens in $q$.

## 3.4   Reward-guided Refinement

As mentioned earlier, the generated task description serves as a guide for creating the compressed prompt by calculating the embedding similarity between the task description and each sentence in the prompt.

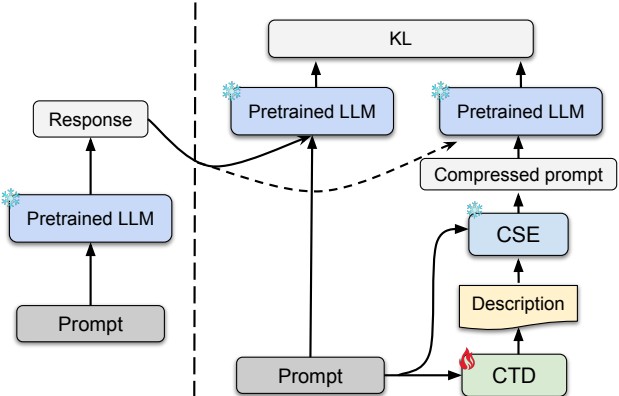

Figure 3: Overview of our proposed reward system for refining CTD with RL. (left) A response is generated by the pre-trained LLM with the complete long input prompt. (right) The CTD and CSE modules generate the compressed prompt. KL divergence between the conditional distribution of the generated response from the long prompt and the compressed prompt is used as the reward signal for the RL.

Consequently, the quality of the compressed prompt is directly dependent on the generated task description. While the dataset $\mathcal{D}_{\text{CTD}}$ generated using the initial prompting pipeline provides a diverse set of $(p, q)$ pairs for training $f_q$, there is no explicit constraint to ensure that $\hat{q}$ generated by the LLM is the most relevant task description for input prompt $p$. To this end, we propose a novel reward function for fine-tuning $f_q$ through cross-entropy reinforcement learning. The reward function is specifically designed to achieve efficient prompt compression by generating a task description $\hat{q}$ that ensures the performance of the compressed prompt is on par with that of the original uncompressed input prompt.

Consider, task-descriptor $f_q$ as a model that performs the action of generating candidate task-description $\hat{q}$ from an input prompt $p$ of a real-world human instruction dataset (e.g. Tulu SFT dataset Ivison et al. (2023)), which is then used to generate the compressed prompt $\mathcal{S} = \phi(\hat{q}, p)$ using Eq. 1. We define the reward function as the KL divergence between the conditional probability of generating the response $r$ from the whole input prompt $p$ and the compressed prompt $\mathcal{S}$ as:

$$R(q_i) = -\text{KL}(P(r \mid \mathcal{S})||P(r \mid p)) \tag{4}$$

where, $r$ is the response from the pre-trained LLM using the whole input prompt $p$ as $r = f_{LLM}(p)$, and $P(r|\mathcal{S})$ is the conditional probability of generating the response $r$ given the compressed prompt $\mathcal{S}$ with a pre-trained LLM $f_{\text{LLM}}$,

$$P(r \mid \mathcal{S}) = f_{\text{LLM}}(\mathcal{S}, r; \theta), \tag{5}$$

and $P(r \mid p)$ is the conditional probability of generating the response $r$ given the original input prompt $p$. The KL divergence measures how much the distribution $P(r \mid \mathcal{S})$ deviates from $P(r \mid p)$, which we use as the reward signal for encouraging the model to generate an informative task description $\hat{q}$ for compressing the prompt. Finally, using $f_q$ as an agent to generate multiple task descriptions $\hat{q}$ (actions), and calculating the corresponding reward, we fine-tune the task descriptor LM with the following loss:

$$\mathcal{L}_{\text{RL}} = -\left[ \sum_{t=1}^{T} \log P(q_{j,t} \mid q_{j,<t}, p; \theta_q) \right], \tag{6}$$

where $q_j$ is the generated task-description with maximal reward from Eq 4. The overall process is illustrated in Figure 3.

### 3.5 Context-aware Sentence Encoder

CSE is the sentence encoder in our method, which captures the information of the surrounding context in the embedding of a sentence. Training CSE consists of two components discussed below: data curation and encoder training.

| Methods | LongBench | | | | | | |
|---|---|---|---|---|---|---|---|
| | SingleDoc | MultiDoc | Summ. | FewShot | Synth. | Code | Tokens |
| Full context | 45.32 | 65.90 | 26.94 | 70.47 | 60.00 | 73.99 | 57.10 |
| *task-aware compression* | | | | | | | |
| Selective-Con. | 29.74 | 43.86 | 23.61 | 57.95 | 22.75 | 45.32 | 1,925 |
| LLMLingua | 20.48 | 37.02 | 21.23 | 55.99 | 10.5 | 43.78 | 1,856 |
| LLMLingua-2 | 33.23 | 52.4 | 24.64 | 46.74 | 26.0 | 59.94 | 1,868 |
| LongLLMLingua | 26.61 | 39.85 | 22.37 | 58.53 | 14.25 | 29.08 | 1,926 |
| CompAct | 19.8 | 40.55 | - | 59.91 | - | - | 831 |
| FAVICOMP | 27.13 | 51.65 | - | 41.63 | - | - | 95 |
| CPC | 43.79 | 62.67 | 25.8 | 67.67 | **54.0** | 56.74 | 1,936 |
| TPC-Base | 44.06 | 62.25 | 25.66 | **70.1** | 53.75 | 59.72 | 1,973 |
| TPC-Large | 42.96 | 58.38 | 25.34 | 69.34 | 52.75 | 58.9 | 1,983 |
| TPC-Huge | **44.7** | **64.02** | **25.84** | 68.91 | **54.0** | **60.04** | 1,969 |
| *task-agnostic compression* | | | | | | | |
| Selective-Con. | 14.72 | 13.08 | 22.04 | 21.09 | 1.0 | 19.3 | 1,756 |
| LLMLingua | 21.02 | 34.79 | 21.32 | 54.57 | 7.00 | **64.62** | 1,870 |
| LLMLingua-2 | 20.18 | 7.00 | 22.86 | 13.02 | 1.34 | 23.3 | 1,866 |
| TPC-Base | 40.70 | 58.23 | 25.66 | 55.95 | 27.38 | 38.36 | 1,934 |
| TPC-Large | 41.59 | 57.59 | **26.26** | 47.86 | **42.0** | 35.25 | 1,978 |
| TPC-Huge | **41.84** | **61.93** | 26.22 | **58.27** | 39.0 | 36.25 | 1,962 |

Table 1: Performance of different methods in task-aware and task-agnostic setups on LongBench.

### 3.5.1 Dataset Curation

To train the context-aware sentence encoder, we require a dataset consisting of tuples containing long contexts, questions, positive sentence, and negative sentences. The context provides all the relevant information necessary to answer the question, positives are sentences within the context that contain partial but not necessarily complete information needed to answer the question, while negative sentences are context sentences that provide no relevant information. While a similar dataset (CQR) was introduced in CPC Liskavets et al. (2025), the simple prompt used in the pipeline did not explicitly ensure that all the information is not present in the positive alone for encouraging understanding of the context. To this end, we introduce a multi-hop question-answer prompt for generating a new dataset of contexts, questions, positive sentence, and negative sentences using Prompt 3 in A.2. The multi-hop question-answer prompt ensures that the generated questions require reasoning over multiple sentences in the context to arrive at an answer, thereby promoting deeper contextual understanding and ensuring that the positive does not contain the entire information required to answer the question. We denote the generated data of $(q, c, pos, \{neg^{i=1\cdots m}\})$ tuples as the Multi-hop Context-Question Relevant dataset (MCQR), which is then used to train the CSE $f_s$.

### 3.5.2 Encoder Training

CSE is trained to learn the context-aware sentence representations by distinguishing the positives and negatives. Specifically, this loss maximizes the similarity between the embeddings of the question and the positive sentence while minimizing the similarity between the question and the embeddings of negative sentences. Similar to CPC Liskavets et al. (2025), we use a contrastive loss along with the MNTP loss BehnamGhader et al. (2024) for more stable training on our newly curated MCQR dataset, with another key modification. Specifically, we introduce two new types of tokens into the tokenizer dictionary: the end-of-sentence token `<end_of_sent>` and the question token `<end_of_question>`. The `<end_of_sent>` token is inserted at the end of each particular sentence in the text, explicitly marking sentence boundaries and enabling the model to process text segments of varying lengths effectively. This token enables the adaptation to different input granularities while maintaining a clear structural understanding of the text. Similarly, the `<end_of_question>` token is appended to the end of each question, signalling its conclusion and helping the model handle questions with diverse syntactic structures. Together, these tokens improve the model's ability to generate embeddings that encode contextual and structural relationships more effectively.

---

**Algorithm 1** Reinforcement Learning for CTD Refinement

---

**Input:** Pre-trained task descriptor $f_q$, candidate task descriptions $\{\hat{q}_i\}$, pre-trained LLM $f_{\text{LLM}}$, input prompt $p$, response $r$
**Output:** Refined task descriptor $f_q$ with updated parameters $\theta_q$
// Generate candidate task descriptions using $f_q$
$\{\hat{q}_i\} = f_q(p; \theta_q)$
// For each $q_i$, construct compressed prompt $\mathcal{S}_i$ using $\phi$
$\mathcal{S}_i = \phi(q_i, p) \quad \forall q_i \in \{\hat{q}_i\}$
// Compute initial response $r$ using $f_{\text{LLM}}$ and the initial non-compressed prompt $p$
$r = f_{\text{LLM}}(p)$
// Evaluate candidate task descriptions based on KL divergence reward
$R(q_i) = -\text{KL}(P(r \mid \mathcal{S}_i) \| P(r \mid p)) \quad \forall q_i \in \{\hat{q}_i\}$
// Optimize $f_q$ using cross entropy RL for the generated task-description with maximal reward $q_j$
$\mathcal{L}_{\text{RL}} = -\left[\sum_{t=1}^{T} \log P(q_{j,t} \mid q_{j,<t}, p; \theta_q)\right]$
Update $\theta_q$ to minimize $\mathcal{L}_{\text{RL}}$
Return refined $f_q$

---

### 3.6 Inference

During inference, we improve the computational and memory efficiencies of deploying TPC by unifying the CTD and CSE encoders. As both tasks utilize the same pretrained encoder, we employ one pre-trained encoder with task-specific LoRA adapters for CTD and CSE. The adapters are activated sequentially to generate $\hat{q}$ from $p$ with CTD, followed by processing $(p, \hat{q})$ with CSE.

### 3.7 Algorithm

Next, we provide the pseudo-code of the overall pipeline of our proposed prompt compression method TPC in Algorithm 3, the pseudo-code for context-relevant task descriptor in Algorithm 2, and CTD refinement with RL in Algorithm 1.

## 4 Experiments

In this section, we discuss the experiments and results of our proposed solution for prompt compression. We begin by presenting the datasets, evaluation protocols, and implementation details. Subsequently, we discuss the main results, ablation studies, and qualitative findings on TPC.

### 4.1 Datasets

**Evaluation datasets.** To evaluate the performance of TPC, we adhere to the experimental protocols outlined in prior research Jiang et al. (2023c) and benchmark our method on **LongBench** Bai et al. (2024) and **ZeroSCROLLS** Shaham et al. (2023) datasets. LongBench features a diverse range of tasks, including both single-document and multi-document QA, summarization, few-shot learning, synthetic and code generation. Similarly, ZeroSCROLLS offers a collection of tasks, including GovReport, SummScreenFD, QMSUM, SQuality, Quality, NarrativeQA, Qasper, Musique, SpaceDigest, BookSumSort, and Tokens.

**Our curated datasets.** As outlined in the Methods section, TPC is trained using two of our curated datasets: CTD and MCQR. The CTD dataset comprises 4,100 context-question pairs specifically designed for training the CTD module. We initialize the seed dataset for training the CTD by combining Tulu-3-sft-

---

**Algorithm 2** Context-relevant Task Descriptor (CTD)

---

**Input:** Prompt $p$, pre-trained LLM $g$, dataset $\mathcal{D}$, training parameters $\theta_q$.
**Output:** Context-relevant Question Generator $f_q$
// Stage 1: Dataset Curation
Initialize empty dataset $\mathcal{D}_{\text{raw}}$
**for** each prompt $p \in \mathcal{D}$ **do**
  Generate raw question $q$ using $g$: $q = g(p; \text{Prompt 1})$
  Append $(p, q)$ to $\mathcal{D}_{\text{raw}}$
**end for**
// Stage 2: Structured Prompt Creation
Initialize empty dataset $\mathcal{D}_{\text{CTD}}$
**for** each $(p, q) \in \mathcal{D}_{\text{raw}}$ **do**
  Generate structured prompt $p_{\text{struct}}$ using $g$: $p_{\text{struct}} = g(p, q; \text{Prompt 2})$
  Append $(p_{\text{struct}}, q)$ to $\mathcal{D}_{\text{CTD}}$
**end for**
// Train Causal Encoder $f_q$
Initialize $f_q$ with pre-trained weights $\theta_q$
**repeat**
  Sample batch $(p_{\text{struct}}, q)$ from $\mathcal{D}_{\text{CTD}}$
  Compute loss:
    $\mathcal{L}_{\text{CTD}} = -\sum_{t=1}^{T} \log P(q_t \mid q_{<t}, p_{\text{struct}}; \theta_q)$
  Update parameters $\theta_q$ using gradient descent
**until** convergence criteria met
Return trained question generator $f_q$

---

**Algorithm 3** Task-Agnostic Prompt Compression (TPC)

---

**Input:** Prompt $p$, context $c$, number of top sentences $k$, context-aware sentence encoder $f_s$, context-relevant task descriptor LM, $f_q$.
**Output:** Compressed prompt $\mathcal{S}$
// Generate task description $\hat{q}$ from prompt $p$ using $f_q$
$\hat{q} = f_q(p; \theta_q)$
// Encode task description $\hat{q}$ and each sentence $s_i$ in context $c$ using sentence encoder $f_s$
$\mathbf{e}_{\hat{q}} = f_s(\hat{q})$, $\{\mathbf{e}_{s_1}, ..., \mathbf{e}_{s_n}\} = f_s(s_1, .., s_n), \forall s_i \in c$
// Compute relevance scores for all sentences in $c$
$\mathcal{R}_i = \mathcal{R}(\mathbf{e}_{\hat{q}}, \mathbf{e}_{s_i}), \forall s_i \in c$
// Select top-$t$ sentences based on relevance scores
$\mathcal{S} = \text{TopK}(\{\mathcal{R}_i \mid s_i \in c\}, k)$
Return compressed prompt $\mathcal{S}$

---

mixture[1], MetaMathFewshot[2], Magicoder-Evol-Instruct-110K[3], consisting of 3,000 prompts. The MCQR dataset includes 9,300 samples, each consisting of a question, a context, a positive and negative sentences.

## 4.2 Evaluation Protocols

We conduct a series of experiments across multiple downstream tasks, following widely accepted protocols on the LongBench and ZeroSCROLLS subsets. Our approach adheres to the standard evaluation protocols established in prior studies Jiang et al. (2023c). Specifically, for summarization tasks, we assess our method by comparing the Rouge scores (Lin, 2004) between the ground truth responses and the outputs generated by the model using compressed prompts. In document QA tasks, the model's responses are evaluated against

---

[1]huggingface.co/datasets/allenai/tulu-3-sft-mixture
[2]huggingface.co/datasets/abacusai/MetaMathFewshot
[3]huggingface.co/datasets/ise-uiuc/Magicoder-Evol-Instruct-110K

| Methods | ZeroSCROLLs | | | | | | | | | | |
| | GovReport | SummScreenFD | QMSUM | SQuality | Quality | NarrativeQA | Qasper | Musique | SpaceDigest | BookSumSort | Tokens |
|---|---|---|---|---|---|---|---|---|---|---|---|
| *task-aware compression* | | | | | | | | | | | |
| Selective-Context | 17.92 | 9.91 | 12.43 | 15.57 | 76.19 | 33.48 | 32.78 | 29.76 | 39.52 | 75.33 | 1,925 |
| LLMLingua | 15.03 | 6.52 | 11.03 | 12.14 | 71.43 | 23.08 | 15.85 | **35.00** | 43.15 | 56.17 | 1,856 |
| LLMLingua-2 | 17.60 | 12.73 | 15.25 | **17.37** | 80.95 | 34.75 | 33.87 | 30.00 | 37.20 | 64.98 | 1,868 |
| LongLLMLingua | 0.00 | 0.00 | 0.00 | 14.45 | 71.43 | 21.6 | 22.81 | 0.00 | 0.00 | 0.00 | 1,926 |
| CompAct | - | - | 11.64 | 13.1 | 66.67 | 17.75 | 4.42 | 20.00 | - | - | 831 |
| FAVICOMP | - | - | 13.14 | 13.0 | 42.86 | 35.86 | 24.05 | 31.76 | - | - | 95 |
| CPC | 18.67 | 12.44 | **17.06** | 16.49 | **100.0** | 45.84 | 41.24 | 29.76 | 54.36 | 80.51 | 1,936 |
| TPC-Base | 18.67 | 12.26 | 15.78 | 17.00 | 90.48 | 44.75 | **41.58** | 29.76 | 53.90 | 80.36 | 2,027 |
| TPC-Large | 18.31 | **13.54** | 16.72 | 16.97 | 90.48 | **46.42** | 32.86 | 34.76 | 57.21 | 80.45 | 1,993 |
| TPC-Huge | **19.30** | 12.14 | 16.38 | 16.64 | **100.00** | 40.75 | 41.35 | 29.76 | **61.42** | **83.33** | 1,941 |
| *task-agnostic compression* | | | | | | | | | | | |
| Selective-Context | 17.94 | 10.71 | 7.96 | 13.14 | 33.33 | 1.14 | 25.58 | 33.33 | 19.31 | **76.5** | 1,756 |
| LLMLingua | 13.75 | 7.37 | 10.15 | 10.56 | 28.57 | 12.56 | 21.50 | 20.0 | 34.53 | 24.54 | 1,870 |
| LLMLingua-2 | 18.24 | **13.09** | 11.07 | 16.33 | 38.10 | 1.11 | **26.74** | **39.76** | 29.49 | 63.63 | 1,866 |
| TPC-Base | **19.72** | 13.07 | **16.94** | **17.29** | **90.91** | 19.97 | 16.54 | 33.11 | 58.73 | 23.00 | 2,032 |
| TPC-Large | 18.48 | 12.30 | 16.02 | **17.29** | **90.91** | **21.6** | **26.7** | 33.29 | 59.51 | 5.24 | 1,978 |
| TPC-Huge | 17.73 | 12.40 | 16.28 | 17.26 | 86.36 | 16.42 | 15.19 | 24.62 | **63.2** | 48.56 | 2,012 |

Table 2: Performance of different methods in task-aware and task-agnostic setups on ZeroSCROLLs.

the ground truth using the F1 score. For code completion tasks, we rely on a textual similarity measure derived from the Levenshtein edit distance (Yujian & Bo, 2007).

We present the main results in two evaluation setups, including task-aware compression and task-agnostic compression. task-aware compression is the most common setup adopted by existing works where the long context is provided with a question, and the compression is performed based on the question and a system prompt. However, this limits its applicability to certain tasks such as summarization and code, where a manual hand-designed and task-specific prompt template is required for the compression. On the contrary, task-agnostic compression does not require such a template or an explicit question to compress the context.We use GPT-4o as the pretrained model to generate final answers. All main results are reported using a compressed prompt limited to 2,000 tokens. We also provide additional extensive evaluations for other generative models (GPT-3.5-turbo, Llama-3.1-8B-Instruct) in A.3. We further assess the scalability of our method with respect to the target compression length by conducting evaluations and comparisons against competitive baselines on the LongBench dataset at a 3,000-token context length. The results are reported in Table 3.

### 4.3 Implementation Details

We use the subset of pile dataset Gao et al. (2020) as the seed dataset, $D_{init}$ in CTD.. We train the $f_q$ with our curated CTD dataset for 2 epochs with an AdamW optimizer, a learning rate of 1.5e-4, and a batch size of 16. This is followed by the reward-guided RL training for 3 iterations with Llama-3.1-8B Dubey et al. (2024) as the pre-trained LLM for the reward function. To obtain questions in the RL stage, we select 16 questions per prompt using a Nucleous Sampling Holtzman et al. with temperature = 0.7 and topP = 0.9. CSE is trained with an AdamW optimizer for 2 epochs, a learning rate of 5e-5, and a batch size of 32. We introduce three versions of TPC, namely Base, Large, and Huge, consisting of Qwen2 Yang et al. (2024) with 0.5B parameters, Llama-3.2-Instruct Dubey et al. (2024) with 1B parameters, and Mistral-Instruct-v0.2 Jiang et al. (2023a) with 7B parameters as the encoder, respectively. We initialize the encoder with the

| Methods | LongBench | | | | | | |
|---|---|---|---|---|---|---|---|
| | SingleDoc | MultiDoc | Summ. | FewShot | Synth. | Code | Avg |
| Full context | 45.32 | 65.90 | 26.94 | 70.47 | 60.00 | 73.99 | 57.10 |
| *Task-aware results* | | | | | | | |
| LLMLingua-1 | 25.94 | 38.20 | 22.27 | 60.01 | 12.25 | 51.13 | 34.97 |
| LongLLMLingua | 30.23 | 43.87 | 23.41 | 59.67 | 20.75 | 31.81 | 34.96 |
| LLMLingua-2 | 38.20 | 57.13 | 25.59 | 53.33 | 35.5 | 63.01 | 45.46 |
| TPC-B | 43.43 | 60.52 | 26.39 | 70.91 | 52.75 | 64.85 | 53.14 |
| TPC-H | 43.73 | 61.02 | 26.59 | 71.66 | 53.45 | 65.74 | 53.70 |
| TPC-L | 43.38 | 59.88 | 25.09 | 69.50 | 53.75 | 65.18 | 52.79 |
| *Task-agnostic results* | | | | | | | |
| LLMLingua-1 | 24.75 | 39.02 | 22.34 | 58.34 | 7.75 | 68.52 | 36.79 |
| LLMLingua-2 | 26.09 | 12.77 | 24.11 | 22.76 | 9.00 | 31.46 | 21.03 |
| TPC-B | 41.69 | 58.81 | 26.12 | 56.49 | 34.81 | 39.71 | 42.94 |
| TPC-L | 41.97 | 59.32 | 26.63 | 48.33 | 42.54 | 38.94 | 42.29 |
| TPC-H | 42.91 | 62.04 | 26.80 | 59.30 | 44.60 | 39.40 | 45.01 |

Table 3: Task-aware and task-agnostic results (3K context length) on LongBench.

| Ablation | Perf. |
|---|---|
| TPC | 41.62 |
| TPC w/o RL | 37.97 |
| TPC w/o SFT and RL | 25.92 |

(a) Ablation on CTD.

| Ablation | Perf. |
|---|---|
| CSE | 51.28 |
| CSE w/o MCQR | 49.7 |
| CSE w/o spec. tokens | 50.23 |

(b) Ablation on CSE.

| Setup | TPC-B | TPC-L | TPC-H | CPC |
|---|---|---|---|---|
| 2K task-aware | 52.59 | 51.28 | 52.92 | 51.78 |
| 3K task-aware | 53.14 | 52.78 | 53.69 | 52.71 |
| 2K task-agnostic | 41.04 | 41.76 | 43.91 | - |
| 3K task-agnostic | 42.93 | 42.28 | 45.01 | - |

(c) Performance vs. context length.

Table 4: Ablation and analysis of different components of TPC on LongBench. In (c), B, L, and H represent Base, Large, and Huge, respectively.

pre-trained weights and fine-tune it with LoRA Hu et al. (2022) of rank 16, and follow the training details from BehnamGhader et al. (2024) to turn the pre-trained causal encoder into bidirectional sentence encoder. All the experiments are conducted on an Nvidia A100 80GB GPU.

## 4.4 Main Results

We present the main results of our experiments on LongBench in Table 1 on both task-aware and task-agnostic compression setups. We perform the evaluations on all three sizes of our model (TPC-Base, TPC-Large, and TPC-Huge). As we find from this table, in the task-aware setup, TPC outperforms all the prior works across the tasks, with up to 3.3 points improvement on individual tasks. While being considerably smaller than the existing SOTA (CPC), our smallest model (TPC-Base) outperforms it on most setups.

Similarly, TPC shows strong performance on the task-agnostic compression setup. Here, only a few of the existing methods can be adopted to the task-agnostic setup, due to the nature of the compression methods. In the task-agnostic setup, TPC outperforms existing methods by a larger margin than the task-aware setup. Specifically, TPC shows up to 35.0 points improvement (on synthetic task) over the existing methods, along with 20.82 points improvement on SingleDoc, and 27.14 points improvement on MultiDoc tasks. Despite having 14× less parameters than our largest model, our Base model achieves strong results, and considerably outperforms existing methods. This gap appears to be a model-family effect rather than a capacity limit: the Qwen-2.5-0.5B backbone in TPC-Base aligns more tightly with our CTD/CSE objectives and yields sharper and less-verbose relevance estimates than the Llama-3-1B backbone in TPC-Large, which is particularly advantageous for task-agnostic compression.

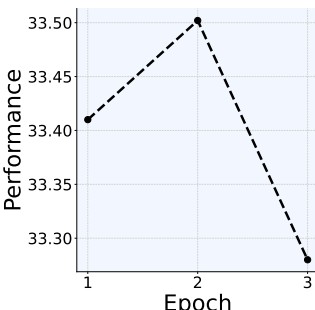 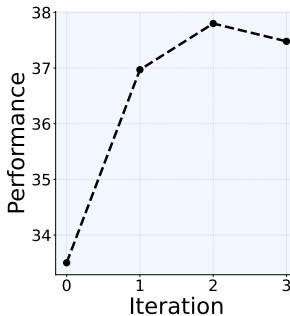

Figure 4: Sensitivity study on CTD training epochs (left) and the number of RL iterations (right). Here, an RL iteration of 0 indicates no reward-guided refinement training.

The results on the ZeroSCROLLs dataset are presented in Table 2. Similar to LongBench, the results are presented on both task-aware and task-agnostic compression setups for the three different model sizes. As we find from this table, on the task-aware setup, TPC outperforms prior works by up to 7.06 points (SpaceDigest). Notably, TPC outperforms existing methods by a larger margin on the task-agnostic setup. Specifically, we observe up to 51.81 points improvement on the Quality task. Additionally, we observe 28.67 and 9.04 points improvements for SpaceDigest and NarrativeQA tasks, respectively. Notably, our base encoder performs comparable to that of the large encoder, surpassing it in a few tasks, possibly due to the relative simplicity of the tasks in ZeroSCROLLs dataset.

While we believe that LongBench and ZeroSCROLLs are the most representative benchmarks, we provide experiments on a broader set of datasets in the appendix.

## 4.5 Ablation

Here, we present ablation and sensitivity experiments on our proposed TPC. All the results in this section are reported as the averaged metrics on the LongBench dataset, with the GPT-4o model utilized as the pre-trained LLM to generate the responses. In Table 4(a) we study the impact of different components of the CTD module, including the supervised fine-tuning and the reward-guided refinement. We also provide results for the non-finetuned CTD model (before applying SFT and RL stages) for comparison. As we find from this experiment, removing the reward-guided refinement module shows a 3.65 point drop in the performance, while removing the ST module shows another 12.05 points drop in the overall performance, showing the importance of both modules in the final performance of the model. Note that RL is dependent on the ST module, and therefore can not be ablated alone.

Next, we ablate our two novel components in CSE, presented in Table 4(b). First, we remove the new multi-hop dataset MCQR and train CSE with CQR instead. This results in a 1.58-point drop in performance, indicating the impact of our curated MCQR dataset over the existing CQR dataset for training a context-aware sentence encoder. Then, we remove the special tokens (<end_of_sent> and <end_of_question>) from the encoder, leading to a 1.05-point drop in performance. These experiments demonstrate the importance of each component in our context-aware sentence encoder.

Next, in Table 4(c), we present additional studies on different constraints for context length, including 2,000 and 3,000 tokens, and compare them with the prior SOTA CPC. From this study, we observe that TPC outperforms CPC under both constraint setups, where TPC-Huge outperforms CPC with a larger margin on the 2K constraint compared to the 3K token constraint.

Next, we perform additional ablation studies to further evaluate the effectiveness of our prompt compression method. To assess the contribution of the CTD module, we replace it with a substantially larger LLM, Mistral-7B-Instruct-v0.2, which we prompt to generate a task description. For scoring the relevance of text segments (sentences) to this task description, we employ CPC. To further evaluate the CSE module, we

| Methods | LongBench | | | | | |
|---|---|---|---|---|---|---|
| | *SingleDoc* | *MultiDoc* | *Summ.* | *FewShot* | *Synth.* | *Code* |
| Full context | 45.32 | 65.90 | 26.94 | 70.47 | 60.00 | 73.99 |
| *task-aware compression* | | | | | | |
| Mistral + CPC | 40.05 | 56.31 | 25.30 | 27.42 | 16.59 | 29.14 |
| LLM2Vec | 4.83 | 1.14 | 12.11 | 0.00 | 0.75 | 17.67 |
| TPC-H | 44.70 | 64.02 | 25.84 | 68.91 | 54.00 | 60.04 |
| *task-agnostic compression* | | | | | | |
| Mistral + CPC | 4.60 | 1.25 | 11.55 | 0.00 | 0.00 | 19.51 |
| LLM2Vec | 1.20 | 0.21 | 2.12 | 3.65 | 3.75 | 5.61 |
| TPC-H | 41.84 | 61.93 | 26.22 | 58.27 | 39.00 | 36.25 |

Table 5: Evaluation of Mistral-generated task descriptions with CPC relevance scoring and LLM2Vec baselines in task-aware and task-agnostic setups on LongBench.

substitute it with LLM2Vec under both task-aware and task-agnostic setups. In the task-aware setup, we use the per-example question provided by LongBench as the task description, and for tasks that lack an explicit question, we use a fixed system prompt. We also use a fixed system prompt for the task-agnostic setup. Additionally, to ensure fair measurement, we leverage the prompt-guided retrieval capability of LLM2Vec by prepending the prefix "Given a user prompt, retrieve relevant passages that address the prompt:" to the task description. Both experiments are conducted on the LongBench benchmark using GPT-4o as the generation model and are summarized in Table 5. As we can see, the simple use of an LLM to generate task descriptions combined with CPC for selecting relevant chunks of the prompt performs reasonably on the task-aware setup, but on the task-agnostic setup it shows very poor results. This indicates the low stability of this approach to prompt distortions during compression. Also, our findings indicate that employing LLM2Vec as a prompt-conditioned encoder is insufficient to achieve effective prompt compression, which demonstrates the importance of the CSE module.

Finally, to further investigate the generalization of our proposed method we perform additional experiments on coherency, intent preservation, contextual information preservation using LLM-as-a-judge. Further details of this experiment are provided in the Appendix A.3.

## 4.6 Sensitivity Analysis

Finally, we investigate two critical hyper-parameters of our method: the number of epochs for training the CTD model and the number of iterations for fine-tuning the CTD using RL. The number of training epochs is crucial to ensure that the task descriptions generated by the CTD model are sufficiently accurate for the subsequent reward-guided fine-tuning stage. As shown in Figure 4 (left), the model's performance drops after 2 epochs of training due to overfitting. The impact of the number of RL iterations is illustrated in Figure 4 (right). Here, iteration 0 represents the baseline performance without RL fine-tuning. As we find from this figure, the performance initially improves with the number of iterations. However, the performance drops after 2 iterations likely due to overfitting to the recent policies.

## 4.7 Latency

In Figure 5, we present the latency evaluations of our method in both task-agnostic and task-aware setups, comparing it with other prompt compression techniques. As shown in the figure, although the latency of our method is slightly higher in the task-agnostic setup due to the overhead introduced by the CTD module, it remains lower than that of most SOTA prompt compression methods, while achieving significantly better performance than existing solutions.

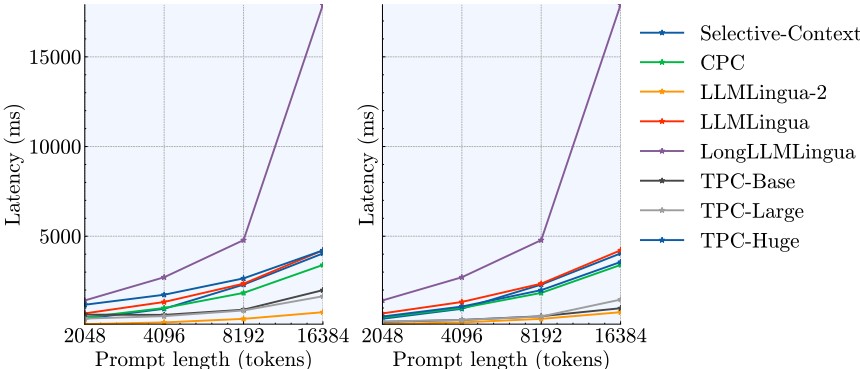

Figure 5: Latency estimates in milliseconds of the competitor methods and the TPC method in task-agnostic setup (left) and TPC method in task-aware setup (right).

## 5  Conclusion

We introduce TPC, a general-purpose prompt compression method that, unlike existing methods doesn't require an explicit question or handcrafted prompt for generating the compressed prompt. Instead, our method uses a task descriptor to generate a context-relevant task description, which is then utilized to find the relevance of each sentence in the prompt to produce the compressed prompt using CSE. Experiments on LongBench and ZeroSCROLLS demonstrate that our approach outperforms existing methods in both task-aware and task-agnostic compression settings. While achieving strong performance, our method is computationally efficient compared to existing SOTA. Our smallest variant with considerably fewer parameters performs on par with the previous SOTA.

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

# A Appendix

## A.1 Qualitative Results

In this section, we first present an example demonstrating the CSE module in action. Figure 9 illustrates the importance of each sentence in the given context, as captured by the `<end_of_sent>` tokens. A darker colour indicates higher importance.

Next, we provide an example of compression in a coding task in Figure 8. First, the CTD module generates the $\hat{q}$ as:

```
CTD obtained task description q̂:
What is the code for the 'test_rw' function in the 'TestSeek' class?
```

Next, the CSE module generates the compression scores using $\hat{q}$. The colour intensity in Figure 8 demonstrates the importance score. The final compressed prompt is visualized in Figure 7.

Finally, we present an example of summarization in Figure 10. Here, the CTD first generates the $\hat{q}$ as:

```
CTD obtained task description q̂:
Write a one-page summary of the report by the government agency on the review process and documentation
of clinical care at VA medical centers.
```

The $\hat{q}$ is then utilized by CSE to generate the compressed prompt as 6.

Finally, we provide examples illustrating the distinction between task-agnostic and task-aware prompt compression below. In the task-agnostic setup, no prior knowledge about the input is assumed; the input is treated as plain text. In contrast, the task-aware setup assumes that the input prompt can be segmented into distinct components, including the system prompt template $I$, question(s) $Q$, and context $C$, which can be compressed.

```
Task agnostic prompt compression:
Please select the answers to the questions presented to you and write out the results in the form of
a table. First, read the text carefully, then give your answer.
Here are the texts:
Once upon a time, dinosaurs lived in the world, it was about 240 million years ago. Among them, ...
stood out.
...
Thus, climate change, provoked by the fall of a meteorite, put an end to the era of dinosaurs.
Questions:
When did dinosaurs live on earth?
What types of dinosaurs are described in the text?
What caused the end of the dinosaur era?

Answer the questions in the form of a table with three columns:   question,   short   answer,
details. Write no more than five sentences in detail for each question
```

```
You are given a report by a government agency. Write a one-page summary of the report.

 Report:
 We found that from October 2013 through March 2017, the five selected VA medical centers required reviews of a total of 148 providers' clin
 ical care after concerns were raised about their care, but officials at these medical centers could not provide documentation to show that a
 lmost half of these reviews were conducted. The medical centers lacked documentation showing that retrospective reviews—which assess the car
 e previously delivered by a provider during a specific period of time— had been conducted for 8 providers after concerns had been raised abo
 ut their clinical care. One medical center lacked documentation showing that reviews had been conducted for another 12 providers after conce
 rns had been raised about their care. In the absence of any documentation, we were unable to identify the types of reviews, if any, that wer
 e conducted for these 12 providers. We also found that the five selected VA medical centers did not always conduct reviews of providers' cli
 nical care in a timely manner. VA concurred with these recommendations and described plans for VHA to revise existing policy and update the
 standardized audit tool used by the networks to include more comprehensive oversight of VA medical centers' reviews of providers' clinical c
 are after concerns have been raised. Now, write a one-page summary of the report.

 Summary:
```

Figure 6: Compressed summarization prompt after taking the top-scored segments.

```
Task aware prompt compression:

I="""
Please select the answers to the questions presented to you and write out the results in the form of
a table. First, read the text carefully, then give your answer.
Here are the texts:
{C}
Questions:
{Q}

Answer the questions in the form of a table with three columns:  question,  short  answer,
details. Write no more than five sentences in detail for each question """

Q="""
When did dinosaurs live on earth?
What types of dinosaurs are described in the text?
What caused the end of the dinosaur era?
"""

C="""
Once upon a time, dinosaurs lived in the world, it was about 240 million years ago. Among them, ...
stood out.
...
Thus, climate change, provoked by the fall of a meteorite, put an end to the era of dinosaurs.
"""
```

## A.2    Dataset curation prompts

```
Prompt 1:
You are tasked with writing user queries based on a long context. Consider the topic of the context
when formulating the query. Queries should be concise and specific. You may also ask more complex
questions, such as requesting specific knowledge extraction from the text, extending the text,
answering questions based on the text, paraphrasing/rewriting the text, or summarizing the text.
Do not include the word "context" in the query.
Long context: {text}
Query:
```

```
Prompt 2:
You are tasked with writing a user query based on a long context. Create a complex template that
integrates this context and a question into a single instruction.
The template must include the placeholders {text} and {question}.
Template examples:
1. You are given a text and a question related to it. Answer the question based on the text.
Question: {question}
Text: {text}
Now answer the question:
2. Can you extend the following block of code: {text}
such that it satisfies the requirement: {question}
Now write the code:
Provide various templates, taking into account the topic of the text and the question.
```

| Methods | Datasets | | | |
|---|---|---|---|---|
| | Krapivin | SummScreenFD | PubMed | Meetingbank |
| *Task-aware evaluation* | | | | |
| CPC | 0.435 | 0**0.2183** | **0.275** | 0.1766 |
| **TPC (ours)** | **0.628** | 0.205 | 0.261 | **0.181** |
| *Task-agnostic evaluation* | | | | |
| CPC with fixed question | 0.439 | 0.201 | 0.270 | 0.172 |
| CPC w/o question | 0.418 | 0.190 | 0.240 | 0.156 |
| **TPC (ours)** | **0.724** | **0.205** | **0.276** | **0.186** |

Table 6: Task-aware and task-agnostic evaluation.

```
Prompt 3:
You are given a long text consisting of numbered sentences. Your task is to generate complex multi-hop
questions about this text, such that answering them requires step-by-step reasoning (in multiple hops).
To achieve this, first, ask a series of sequential factual questions and identify the corresponding
sentences that contain the answers to these questions. Then, formulate a final question that can
only be answered by combining the information from the previously generated questions and answers.
Additionally, specify which sentences (by their numbers) contain the information necessary to answer
the final question.

Toy example:  [[1]] John is married to Mary.  [[2]] They've decided to spend their marriage
anniversary in Spain. [[3]] Mary was afraid that their two small children, Jody and Sue, were too
small for a flight. [[4]] That's why she asked her elder sister Jane to look after them.
Questions:
Question 1: Who is John married to?
Answer 1: John is married to Mary, as stated in [[1]].
Question 2: How many children does Mary have?
Answer 2: Mary has two children, as stated in [[3]].

Combining the questions to create a multi-hop question:
1. John is married to Mary, as stated in [[1]].
2. Mary has two children, as stated in [[3]].
Final question: How many children does John have?
Necessary sentences: [[1]], [[3]]

Now, solve this example:
{text}
Questions:
```

## A.3 Experiments

To further highlight the efficiency of our method against CPC, we conduct experiments on 4 additional out-of-distribution datasets. The results are presented in Table 6. As evident from the results, while being 7 times smaller (1B parameters) than CPC (7B parameters), TPC outperforms it on keyword extraction (Krapivin), and summarizing meetings (meetingbank), and performs as per on the PubMed and SummScreenFD datasets. More importantly, unlike CPC, TPC does not require any handcrafted prompts, explicit question format, or prompt template to work on these new tasks. Furthermore, TPC outperforms CPC with a larger margin in the prompt-agnostic setup. These results emphasize the effectiveness and generalization of the task-agnostic nature of TPC.

Next, we compare our method with a concurrent work, DAC (Zhao et al., 2025), on a subset of the datasets on which they evaluated their method. The results are presented in Table 7. As we find from this table, our method considerably outperforms DAC on most datasets, with DAC performing marginally better on only 3 datasets.

Next, we compare our method against LanguaShrink Ma et al. (2024) under the same evaluation protocol described in their paper. The results, summarized in Table 8, indicate that our approach marginally outperforms LanguaShrink on 4 of the 6 LongBench tasks.

```
Please complete the code given below.
#! /usr/bin/env python
"""Test for the sndfile class."""
# XXX: there is a lot to refactor here
class TestSndfile(TestCase):
    def test_basic_io(self):
        """ Check open, close and basic read/write"""
        # dirty !
    def test_basic_io_fd(self):
        """ Check open from fd works"""
    def test_raw(self):
        rawname = join(TEST_DATA_DIR, 'test.raw')
        """Check float32 write/read works"""
        """Check 32 bits pcm write/read works"""
        """Check 16 bits pcm write/read works"""
        self._test_read_write(np.int16)
    def _test_read_write(self, dtype):
    #def test_supported_features(self):
        """Check for bad arguments."""
        """ Try to seek really far."""
                                e.__class__)
class TestSeek(TestCase):
    def test_simple(self):
        ofilename = join(TEST_DATA_DIR, 'test.wav')
        # Open the test file for reading
    def test_rw(self):
        """Test read/write pointers for seek."""
        ofilename = join(TEST_DATA_DIR, 'test.wav')
 Next line of code:
```

Figure 7: Compressed coding prompt after taking the top-scored segments.

| Methods | Datasets | | | | | | | | |
| --- | --- | --- | --- | --- | --- | --- | --- | --- | --- |
| | NarrativeQA | Qasper | MultifieldQA | HotpotQA | 2WikiMQA | Musique | GovReport | MultiNews | QMSum |
| DAC | **24.85** | 33.46 | 40.12 | 42.37 | 39.57 | 22.44 | 30.40 | **25.77** | **25.95** |
| TPC (ours) | 23.72 | **37.75** | **50.25** | **45.06** | **53.87** | **26.78** | **31.15** | 25.45 | 21.82 |

Table 7: Comparison of DAC and TPC across nine datasets.

## A.4 Generalization to alternative LLM models

To further demonstrate the generalizability of our method to alternative models, we conduct additional evaluations on GPT-3.5-turbo and Llama-3.1-8B-Instruct. The results are presented in Tables 8 and 9.

## A.5 Out-of-Domain Generalization evaluation using LLM-as-a-Judge

We conduct additional evaluations to assess the generalization capability of our prompt compression method, measuring quality across multiple dimensions using an LLM-as-a-Judge. Specifically, we evaluate the compressed prompts along the following dimensions:

- Coherency: We assess how consistent the compressed prompt is with the original version by asking an LLM to rate its coherence on a scale from 1 to 5.
- Intent preservation (IR): We evaluate how well the compressed prompt retains the user's original intent, as expressed in the uncompressed prompt
- Contextual information preservation (CIP): We assess the extent to which the compressed prompt enables the LLM to generate a response similar to what it would produce when given the original prompt

| Model | GPT-3.5-turbo (context length 2K) | | | | | | |
|---|---|---|---|---|---|---|---|
| | SingleDoc | MultiDoc | Summ. | FewShot | Synth | Code | Avg |
| *Task-aware setup* | | | | | | | |
| LLMLingua | 22.40 | 32.10 | 24.50 | 61.20 | 10.40 | 56.80 | 34.60 |
| LLMLingua-2 | 29.80 | 33.10 | 25.30 | 66.40 | 21.30 | 58.90 | 39.10 |
| LanguaShrink | 42.10 | 54.30 | 26.30 | 62.30 | 33.00 | 58.40 | 46.10 |
| TPC-B | 42.54 | 49.56 | 24.80 | 67.79 | 55.63 | 55.96 | 49.38 |
| TPC-L | 42.92 | 47.95 | 24.88 | 68.17 | 56.13 | 57.70 | 49.62 |
| TPC-H | 43.92 | 50.28 | 25.43 | 68.87 | 55.63 | 59.53 | 50.61 |
| *Task-agnostic setup* | | | | | | | |
| LLMLingua | 19.02 | 26.84 | 19.55 | 50.72 | 6.25 | 38.99 | 26.90 |
| LLMLingua-2 | 18.38 | 6.86 | 22.09 | 20.78 | 0.78 | 31.11 | 16.67 |
| TPC-L | 37.65 | 42.99 | 23.74 | 59.36 | 30.75 | 37.38 | 38.64 |
| TPC-B | 40.67 | 47.09 | 23.44 | 60.98 | 31.66 | 39.64 | 40.58 |
| TPC-H | 42.93 | 47.47 | 23.30 | 61.47 | 43.38 | 40.93 | 43.25 |

Table 8: Performance of GPT-3.5-turbo on LongBench (context length 2K) across task-aware and task-agnostic setups.

| Model | Llama-3.1-8B-Instruct (context length 2K) | | | | | | |
|---|---|---|---|---|---|---|---|
| | SingleDoc | MultiDoc | Summ. | FewShot | Synth | Code | Avg |
| Full context | 44.83 | 46.10 | 24.14 | 34.48 | 53.26 | 13.27 | 36.01 |
| *Task-aware setup* | | | | | | | |
| LLMLingua-1 | 13.86 | 17.09 | 20.42 | 51.74 | 4.94 | 38.73 | 24.46 |
| LLMLingua-2 | 19.17 | 24.65 | 21.31 | 54.39 | 12.93 | 33.96 | 27.73 |
| TPC-B | 36.56 | 42.48 | 25.05 | 65.29 | 48.85 | 41.81 | 43.34 |
| TPC-L | 35.26 | 39.91 | 24.89 | 65.15 | 49.13 | 41.85 | 42.70 |
| TPC-H | 36.68 | 43.37 | 25.38 | 66.38 | 47.28 | 42.07 | 43.52 |
| *Task-agnostic setup* | | | | | | | |
| LLMLingua-1 | 10.95 | 9.68 | 20.30 | 48.56 | 2.33 | 34.67 | 21.08 |
| LLMLingua-2 | 15.85 | 4.37 | 17.02 | 19.58 | 1.68 | 23.85 | 13.72 |
| TPC-B | 33.40 | 33.41 | 23.88 | 41.46 | 17.02 | 32.60 | 30.29 |
| TPC-L | 31.93 | 33.42 | 22.92 | 25.94 | 22.79 | 31.02 | 28.00 |
| TPC-H | 26.11 | 17.49 | 20.52 | 45.24 | 33.72 | 34.72 | 29.63 |

Table 9: Performance of Llama-3.1-8B-Instruct on LongBench (context length 2K) across task-aware and task-agnostic setups.

The evaluations are conducted on the following out-of-distribution datasets:

- Self-Instruct Wang et al. (2023), which includes complex prompts generated by LLM
- P3 Sanh et al. (2022), a subset of crowd-sourced instructions and demonstrations sampled from the Public Pool of Prompts (P3) dataset
- UltraChat Ding et al. (2023), a subset of Long instruction-following prompts curated for UltraLM. Examples include multi-paragraph instructions, simulated conversations, and educational tasks
- SystemCheck Mu et al. (2025). A subset of the dataset containing generated user queries targeting 14 test-only system prompts with irrelevant in-context task demonstrations
- A subset of OpenOrca Mukherjee et al. (2023), extended OpenAI prompts (via ShareGPT + FLAN Wei et al.), many with long, deeply nested reasoning tasks

| Dataset | Method | Coherency | IR | CIP | Winrate |
|---|---|---|---|---|---|
| *Comparison with LLMLingua-2* | | | | | |
| Financial News Prompts | LLMLingua-2 | 2.5 | 2.5 | 1.6 | 0.6 |
| Financial News Prompts | TPC | 3.6 | 4.7 | 3.4 | 0.4 |
| Self-Instruct | LLMLingua-2 | 2.2 | 2.5 | 1.8 | 0.2 |
| Self-Instruct | TPC | 3.6 | 2.8 | 2.4 | 0.8 |
| P3 | LLMLingua-2 | 3.2 | 3.6 | 2.9 | 0.4 |
| P3 | TPC | 2.9 | 2.8 | 2.3 | 0.6 |
| UltraChat | LLMLingua-2 | 2.4 | 2.9 | 2.3 | 0.1 |
| UltraChat | TPC | 3.7 | 3.4 | 2.7 | 0.9 |
| SystemCheck | LLMLingua-2 | 1.8 | 2.8 | 2.4 | 0.1 |
| SystemCheck | TPC | 3.6 | 3.1 | 2.3 | 0.9 |
| OpenOrca | LLMLingua-2 | 2.8 | 3.1 | 2.5 | 0.1 |
| OpenOrca | TPC | 3.4 | 3.3 | 2.4 | 0.9 |
| *Comparison with SelectiveContext* | | | | | |
| Financial News Prompts | SelectiveContext | 2.6 | 2.6 | 1.8 | 0.1 |
| Financial News Prompts | TPC | 3.6 | 4.7 | 3.4 | 0.9 |
| Self-Instruct | SelectiveContext | 2.7 | 3.1 | 2.1 | 0.1 |
| Self-Instruct | TPC | 3.6 | 2.8 | 2.4 | 0.9 |
| P3 | SelectiveContext | 3.2 | 3.1 | 2.7 | 0.1 |
| P3 | TPC | 2.9 | 2.8 | 2.3 | 0.9 |
| UltraChat | SelectiveContext | 2.3 | 2.4 | 1.7 | 0.0 |
| UltraChat | TPC | 3.7 | 3.4 | 2.7 | 1.0 |
| SystemCheck | SelectiveContext | 1.9 | 2.4 | 1.6 | 0.0 |
| SystemCheck | TPC | 3.6 | 3.1 | 2.3 | 1.0 |
| OpenOrca | SelectiveContext | 2.7 | 3.0 | 2.3 | 0.2 |
| OpenOrca | TPC | 3.4 | 3.3 | 2.4 | 0.8 |
| *Comparison with LLMLingua-1* | | | | | |
| Financial News Prompts | LLMLingua-1 | 1.7 | 1.6 | 1.2 | 0.0 |
| Financial News Prompts | TPC | 3.6 | 4.7 | 3.4 | 1.0 |
| Self-Instruct | LLMLingua-1 | 2.2 | 2.4 | 2.0 | 0.0 |
| Self-Instruct | TPC | 3.6 | 2.8 | 2.4 | 1.0 |
| P3 | LLMLingua-1 | 1.9 | 2.6 | 1.8 | 0.0 |
| P3 | TPC | 2.9 | 2.8 | 2.3 | 1.0 |
| UltraChat | LLMLingua-1 | 1.5 | 1.7 | 1.3 | 0.0 |
| UltraChat | TPC | 3.7 | 3.4 | 2.7 | 1.0 |
| SystemCheck | LLMLingua-1 | 1.5 | 1.6 | 1.5 | 0.0 |
| SystemCheck | TPC | 3.6 | 3.1 | 2.3 | 1.0 |
| OpenOrca | LLMLingua-1 | 1.8 | 2.2 | 1.7 | 0.0 |
| OpenOrca | TPC | 3.4 | 3.3 | 2.4 | 1.0 |

Table 10: Comparison of TPC against LLMLingua-2, SelectiveContext, and LLMLingua-1 across multiple datasets using LLM-as-a-Judge.

These experiments are conducted on a randomly sampled subset of 200 examples from each dataset. As shown in the Table 10, our method consistently achieves higher scores than existing approaches. Additionally, for each original instruction prompt from a given dataset, we directly ask the GPT-4.1 LLM to choose between the compressed prompt generated by our method and that produced by a competing method. We define the win rate of compressed prompt A over B as the number of times A is preferred over B, divided by the total number of samples from that dataset. A higher win rate thus reflects a more effective prompt compression strategy. According to the evaluation results, GPT-4.1 consistently prefers prompts compressed by our method over those produced by LLMLingua-2. We also observe a substantial performance advantage over other task-agnostic prompt compression baselines.

```
Please complete the code given below.  <end_of_sent>
#! /usr/bin/env python <end_of_sent>
# Last Change: Sun Dec 14 07:00 PM 2008 J <end_of_sent>
"""Test for the sndfile class.""" <end_of_sent>
from os.path import join, dirname <end_of_sent>
import os <end_of_sent>
import sys <end_of_sent>
from numpy.testing import TestCase, assert_array_equal, dec <end_of_sent>
import numpy as np <end_of_sent>
from audiolab import Sndfile, Format, available_encodings, available_file_formats <end_of_sent>
from testcommon import open_tmp_file, close_tmp_file, TEST_DATA_DIR <end_of_sent>
_DTYPE_TO_ENC = {np.float64 : 'float64', np.float32: 'float32',  <end_of_sent>
                 np.int32: 'pcm32', np.int16: 'pcm16'} <end_of_sent>
# XXX: there is a lot to refactor here <end_of_sent>
class TestSndfile(TestCase): <end_of_sent>
...
...
...
        a = Sndfile(ofilename, 'r') <end_of_sent>
        a.seek(-buffsize, 1) <end_of_sent>
        assert_array_equal(buff, buff2) <end_of_sent>
        a.close() <end_of_sent>
        # Now, read some frames, go back, and compare buffers <end_of_sent>
        # (check whence == 2 == SEEK_END) <end_of_sent>
        a = Sndfile(ofilename, 'r') <end_of_sent>
        buff = a.read_frames(nframes) <end_of_sent>
        a.seek(-buffsize, 2) <end_of_sent>
        buff2 = a.read_frames(buffsize) <end_of_sent>
        assert_array_equal(buff[-buffsize:], buff2) <end_of_sent>
    def test_rw(self): <end_of_sent>
        """Test read/write pointers for seek.""" <end_of_sent>
        ofilename = join(TEST_DATA_DIR, 'test.wav') <end_of_sent>
 Next line of code: <end_of_sent>
```

Figure 8: CSE segment scores colormap for coding prompt. Darker colours indicate higher scores.

## A.6 Licenses

We used pre-trained models and open-source datasets. In terms of models, we use models from the Qwen-2.5 family and Mistral family, which are licensed under the APACHE-2.0 license. We also used the LLama-3.2 model, which is licensed under the "LLAMA 3.2 COMMUNITY LICENSE AGREEMENT". All the models used are available for research purposes.

As our training dataset, we used "allenai/tulu-3-sft-mixture", which is licensed under the "Open Data Commons License Attribution family", as well as "abacusai/MetaMathFewshot" and "ise-uiuc/Magicoder-Evol-Instruct-110K", which are licensed under the APACHE-2.0 license. We assume that these data follow the well-known HHH paradigm. Therefore, taking data from these sources should not introduce potentially malicious behaviour into our models. We also used the Pile dataset, which is licensed under the MIT license. We conduct manual checks on subsamples of the datasets to verify the cleanliness and safety of the data. In addition, we provide full details on how we created the datasets in Section 3.5.1.

We evaluate our method on benchmarks such as MIT-licensed LongBench and MIT-licensed ZeroScrolls. All aforementioned models, datasets, and benchmarks are publicly available and imply research use. Their prescribed terms of use are hereby aligned with our research work.

Figure 9: Visualization of the operation of the compression model with special tokens. Each <end_of_sent> token is highlighted with an intensity proportional to its importance in terms of answering the question.

Figure 10: CSE segment scores colormap for summarization prompt. Darker colours indicate higher scores.

