# OpenReview forum: "Task-agnostic Prompt Compression with Context-aware Sentence Embedding and Reward-guided Task Descriptor"
_TMLR — Accepted by TMLR_

### Review · Reviewer_Kcp7 · 2025-10-05

**Summary Of Contributions:**

The paper introduces Task-agnostic Prompt Compression (TPC), a framework for reducing prompt length in LLMs without relying on explicit user queries or handcrafted templates. Unlike most prior compression methods that depend on task-specific signals, TPC leverages a context-relevant task descriptor (a lightweight LM) to generate a task description directly from the input prompt. This description is then used with a context-aware sentence encoder to assess sentence relevance and construct the compressed prompt. To improve the quality of task descriptions, the model incorporates reinforcement learning with KL-divergence reward, ensuring that compressed prompts preserve essential information needed for accurate LLM responses. The authors evaluate TPC on two benchmarks (LongBench and ZeroSCROLLS) under both task-aware and task-agnostic setups, demonstrating that their approach outperforms the baselines.

**Strengths:**
- TPC addresses a key limitation of prior methods (e.g., CPC) by eliminating the need for explicit questions or handcrafted templates.
- The RL fine-tuning of the CTD is elegant and directly ties compression to downstream performance.
- The multi-hop dataset curation for CSE promotes deeper contextual reasoning

**Weakness**
- The paper lacks some key comparisons and ablation experiments to support the claims.

**Audience:**

Yes

**Audience Explanation:**

TMLR's audience would likely find this work appealing due to its advances in prompt efficiency, which is a key challenge in LLM deployment. However, the submission would be more impactful if the raised concerns are addressed.

**Broader Impact Concerns:**

I think no major ethical concerns arise from this work, as it focuses on efficiency improvements for LLMs without introducing novel risks.

**Claims And Evidence:**

No

**Claims Explanation:**

While the paper presents a good approach to task-agnostic prompt compression, some key limitations warrant attention:

- **Insufficient Empirical Evidence for the CSE Over Existing Methods**
In Section 2.2, the authors claim that models like LLM2Vec (BehnamGhader et al., 2024) "excel at sentence-level representation but lack context awareness," which motivates the development of CSE as a key contribution. However, LLM2Vec supports instruction-based embeddings (e.g., prepending prompts like "Retrieve semantically similar text"). Similarly, MAGNET [1] (not cited in the paper) generates prompt-conditioned sentence embeddings (check Section 4.2 of [1]). The paper doesn't include experiments comparing CSE to these baselines (e.g., ablating TPC with LLM2Vec or MAGNET as the encoder), which undermines the novelty or necessity of CSE.

- **Discrepancies in Baseline Results and Limited Evaluation Across Downstream LLMs**
The reported task-agnostic results for LLMLingua-2 in Table 1 are substantially lower than those in the original paper. Is this because a different pretrained LLM is used for generating the final answer? To demonstrate TPC's generalizability (LLM-agnosticism) evaluations across multiple models are essential, alongside explanations for baseline mismatches.

- **Lack of Analysis on Compression Ratios and Uncompressed Baselines in Task-Agnostic Settings**
Task-agnostic results are reported at a fixed ~2k token limit. There is no analysis for varying compression ratio for the task-agnostic setup, despite its focus on broad applicability. Additionally, uncompressed prompt baselines (pretrained LLM performance without compression) are not reported.

- **Missing Comparisons to Recent Task-Agnostic Baselines**
For the task-agnostic setting, the paper doesn't compare their approach with methods like DAC [2], Selection-p [3], and LanguaShrink [4].

[1] Khosla et al., "MAGNET: Augmenting Generative Decoders with Representation Learning and Infilling Capabilities," ACL 2025.

[2] Zhao el al., "DAC: A Dynamic Attention-aware Approach for Task-Agnostic Prompt Compression," ACL 2025.

[3] Chung et al., "Selection-p: Self-Supervised Task-Agnostic Prompt Compression for Faithfulness and Transferability," EMNLP 2024.

[4] Liang et al., "LanguaShrink: Reducing Token Overhead with Psycholinguistics," ArXiv 2024.

**Requested Changes:**

I recommend the following additions to secure my recommendation for acceptance:
- Replace CSE with off-the-shelf LLM2Vec/MAGNET encoders in the TPC pipeline and report LongBench/ZeroSCROLLS performance
- Expand evaluations in Tables 1-2 to include additional LLMs (e.g., GPT-3.5-Turbo, Llama-3-8B).
- In Tables 1-2 and Figure 1, report uncompressed baselines.
- Extend ablation in Table 3(c) to task-agnostic setting.
- Add comparison to stronger task-agnostic prompt compression methods like [2,3,4].
- Briefly explain why TPC-Base outperform TPC-Large on most tasks.

I also recommend the following minor adjustments:
- Fix the title of the vertical axis in Figure 1. It should be “task-agnostic” and task-aware.”
- Cite recent works like [1,2,3,4] in Section 2.

[1] Khosla et al., "MAGNET: Augmenting Generative Decoders with Representation Learning and Infilling Capabilities," ACL 2025.

[2] Zhao el al., "DAC: A Dynamic Attention-aware Approach for Task-Agnostic Prompt Compression," ACL 2025.

[3] Chung et al., "Selection-p: Self-Supervised Task-Agnostic Prompt Compression for Faithfulness and Transferability," EMNLP 2024.

[4] Liang et al., "LanguaShrink: Reducing Token Overhead with Psycholinguistics," ArXiv 2024.

---

> ### Author Response · Authors · 2025-11-02
>
> We would like to thank the reviewer for their thoughtful feedback and comments. Below we have answered the questions and reflected the changes in the revised manuscript in *blue*. Please don't hesitate to let us know if anything was unclear or you required additional clarification.
>
>
> > Insufficient empirical evidence for the CSE over existing methods
>
>
> To address the reviewer’s concern, we conducted additional experiments on an open-source method (LLM2Vec). We used the following setup:
>
> *"Given a user prompt, retrieve relevant passages that address the prompt:"*
>
> In this experiment, the text is first segmented and the most relevant segments are selected according to the user query. For tasks where a user query is provided, we use it together with the system prompt; for tasks like summarization, we use the system prompt, which generally contains a description of the overall task for all benchmark examples.
>
> We evaluated LLM2Vec in both **task-aware** and **task-agnostic** setups. The results are shown below:
>
> | Method        | SingleDoc | MultiDoc | Summarization | FewShot | Synth | Code  |
> | ------------- | --------- | -------- | ------------- | ------- | ----- | ----- |
> | Task-Agnostic | 1.20       | 0.21     | 2.12          | 3.65    | 3.75  | 5.61  |
> | Task-Aware    | 4.83      | 1.14     | 12.11         | 0.00     | 0.75  | 17.67 |
>
>
> We have now added these experiments to page 13 of the revised paper.
>
> > Lack of analysis on compression ratios
>
> To address the reviewer’s concern, we have now conducted additional evaluations with **3k context length** and full context on GPT-4o, including measurements for competing methods. As we observe from the table below, the performance slightly decreases when moving from 3k to 2k context length; however, the drop is not significant, and the model still preserves key information. Interestingly, in some domains such as Q&A, performance even improves under higher compression, indicating that irrelevant information is effectively removed. These new experiments are now added to page 11 of the revised paper.
>
> **Task-aware results (3k context length):**
>
> | Base Model    | SingleDoc | MultiDoc | Summarization | FewShot | Synth | Code   | Avg     |
> | ------------- | --------- | -------- | ------------- | ------- | ----- | ------ | ------- |
> | LLMLingua-1   | 25.94   | 38.20  | 22.27         | 60.01 | 12.25 | 51.13  | 34.97 |
> | TPC-B         | 43.43   | 60.52   | 26.39         | 70.91   | 52.75 | 64.85 | 53.14 |
> | TPC-H         | 43.73    | 61.02   | 26.59         | 71.66   | 53.45 | 65.74 | 53.70 |
> | TPC-L         | 43.38   | 59.88  | 25.09         | 69.50 | 53.75 | 65.18 | 52.79 |
> | LLMLingua-2   | 38.20   | 57.13  | 25.59         | 53.33 | 35.50 | 63.01 | 45.46 |
> | LongLLMLingua | 30.23   | 43.87    | 23.41         | 59.67 | 20.75 | 31.81  | 34.96 |
>
> **Task-agnostic results (3k context length):**
>
> | Base Model  | SingleDoc | MultiDoc | Summarization | FewShot | Synth | Code   | Avg     |
> |-|-|-|-|-|-|-|-|
> | LLMLingua-1 | 24.75   | 39.02  | 22.34         | 58.34 | 7.75  | 68.52 | 36.79 |
> | LLMLingua-2 | 26.09     | 12.77    | 24.11       | 22.76 | 9.00     | 31.46 | 21.03 |
> | TPC-B       | 41.69     | 58.81  | 26.12         | 56.49   | 34.81 | 39.71  | 42.94 |
> | TPC-L       | 41.97   | 59.32  | 26.63       | 48.33   | 42.54 | 38.94 | 42.29 |
> | TPC-H       | 42.91     | 62.04    | 26.80          | 59.30    | 44.60  | 39.40   | 45.01 |
>
> Based on these results, we propose updating **Table 3(c)** with the following values:
>
> | Setup            | TPC-B | TPC-L  | TPC-H |
> | ---------------- | ----- | ------ | ----- |
> | Task-aware 2k    | 52.59 | 51.28  | 52.92 |
> | Task-aware 3k    | 53.14 | 52.78  | 53.69 |
> | Task-agnostic 2k | 41.04 | 41.76 | 43.91 |
> | Task-agnostic 3k | 42.93 | 42.28  | 45.01 |
>
> Values for **task-agnostic 2k** are taken from Table 1.
>
> Also, we provide the results for uncompressed prompts using GPT-4o as a generation model and Llama-3.1-8B-Instruct:
>
> | base_model                         |   singledoc |   multidoc |   summarization |   fewshot |   synth |   code |     avg |
> |-----------------------------------|------------:|-----------:|----------------:|----------:|--------:|-------:|--------:|
> | GPT-4o |       45.32 |    65.90 |         26.94 |   70.47 |      60 | 73.99 | 57.10 |
> | Llama-3.1-8B-Instruct |       44.83 |    46.10 |         24.14 |     34.48 |   53.26 | 13.27 | 36.01 |

---

> > ### Author Response · Authors · 2025-11-02
> >
> > > Discrepancies in Baseline Results and Limited Evaluation Across Downstream LLMs The reported task-agnostic results for LLMLingua-2 in Table 1 are substantially lower than those in the original paper. Is this because a different pretrained LLM is used for generating the final answer? To demonstrate TPC's generalizability (LLM-agnosticism) evaluations across multiple models are essential, alongside explanations for baseline mismatches.
> >
> >
> > The reason for the discrepancy comes from the change in the pre-trained LLM. LLMLingua* evaluated their models using **GPT-3.5-turbo**, which is considerably outdated. Therefore, we evaluated the existing methods and our proposed method on the latest **GPT-4o** model. To further address the reviewer’s concern, we also report the results obtained with **GPT-3.5-turbo**, shown below for our method (the author can then compare them to the results of LLMLingua from the official paper).
> >
> > Additionally, we conduct experiments under two setups: **task-aware prompt compression** and **task-agnostic prompt compression**, while all existing methods reported only the results for the first setup.
> >
> > In the **task-aware** setup, the model compresses only a selected part of the context, which is then embedded into a predefined template consisting of a system prompt, a question, and other structural elements before being passed to the generative model for evaluation.
> >
> > For example:
> >
> > ```
> > You are a smart assistant. You will be given several texts which are in mixed order. Provide the correct order of the texts’.
> >
> > Text 1: [[context to compress…]]
> > Text 2: [[context to compress…]]
> > …
> > Text N: [[context to compress…]]
> >
> > Your response:
> > ```
> >
> > Here, only the text inside the brackets is subject to compression.
> >
> > In the **task-agnostic** setup, the model is asked to compress the *entire* prompt—including the instructions and the context—so that it can be directly passed to the LLM without any additional templating or manual modifications:
> >
> > ```
> > [[You are a smart assistant. You will be given several texts which are in mixed order. Provide the correct order of the texts’.
> >
> > Text 1: context to compress…
> > Text 2: context to compress…
> > …
> > Text N: context to compress…
> >
> > Your response:]]
> > ```
> >
> > In their experiments, LLMLingua-* used the first (task-aware) setup. However, we argue that this does not represent a truly fair prompt compression scenario, since it relies on external manual structuring. For completeness, we report results for both setups in our paper.
> >
> > **Task-aware setup (context length 2k, GPT-3.5-turbo):**
> >
> > | Model | SingleDoc | MultiDoc | Summarization | FewShot | Synth |  Code |   Avg |
> > | :---- | --------: | -------: | ------------: | ------: | ----: | ----: | ----: |
> > | TPC-L |     42.92 |    47.95 |         24.88 |   68.17 | 56.13 | 57.70 | 49.62 |
> > | TPC-B |     42.54 |    49.56 |         24.80 |   67.79 | 55.63 | 55.96 | 49.38 |
> > | TPC-H |     43.92 |    50.28 |         25.43 |   68.87 | 55.63 | 59.53 | 50.61 |
> >
> > **Task-agnostic setup (context length 2k, GPT-3.5-turbo):**
> >
> > | Model       | SingleDoc | MultiDoc | Summarization | FewShot | Synth |  Code |   Avg |
> > | :---------- | --------: | -------: | ------------: | ------: | ----: | ----: | ----: |
> > | LLMLingua-1 |     19.02 |    26.84 |         19.55 |   50.72 |  6.25 | 38.99 | 26.90 |
> > | LLMLingua-2 |     18.38 |     6.86 |         22.09 |   20.78 |  0.78 | 31.11 | 16.67 |
> > | TPC-L       |     37.65 |    42.99 |         23.74 |   59.36 | 30.75 | 37.38 | 38.64 |
> > | TPC-B       |     40.67 |    47.09 |         23.44 |   60.98 | 31.66 | 39.64 | 40.58 |
> > | TPC-H       |     42.93 |    47.47 |         23.30 |   61.47 | 43.38 | 40.93 | 43.25 |
> >
> >
> > Also, we provide the results for Llama3.1-8B-Instruct for the context length 2k for both setups.
> > Task-aware setup:
> > |base_model|singledoc|multidoc|summarization|fewshot|synth|code|avg|
> > |-|-|-|-|-|-|-|-|
> > |LLMLingua-1|13.86|17.09|20.42|51.74|4.94|38.73|24.46|
> > |LLMLingua-2|19.17|24.65|21.31|54.39|12.93|33.96|27.73|
> > |TPC-B|36.56|42.48|25.05|65.29|48.85|41.81|43.34|
> > |TPC-L|35.26|39.91|24.89|65.15|49.13|41.85|42.70|
> > |TPC-H|36.68|43.37|25.38|66.38|47.28|42.07|43.52|
> >
> > Task agnostic setup:
> > |base_model|singledoc|multidoc|summarization|fewshot|synth|code|avg|
> > |-|-|-|-|-|-|-|-|
> > |LLMLingua-1|10.95|9.68|20.30|48.56|2.325|34.67|21.08|
> > |LLMLingua-2|15.85|4.37|17.02|19.58|1.68|23.85|13.72|
> > |TPC-B|33.40|33.41|23.88|41.46|17.02|32.60|30.29|
> > |TPC-L|31.93|33.42|22.92|25.94|22.79|31.02|28.00|
> > |TPC-H|26.11|17.49|20.52|45.24|33.72|34.72|29.63 |
> >
> >
> > These results show that TPC consistently maintains strong performance across different LLMs, confirming its generalizability and robustness while explaining discrepancies with prior baselines. These experiments are now added to page 21 of the paper.

---

> > > ### Author Response · Authors · 2025-11-02
> > >
> > > > Missing comparisons to recent task-agnostic baselines for the task-agnostic setting, the paper doesn't compare their approach with methods like DAC [2], Selection-p [3], and LanguaShrink [4].
> > >
> > >
> > > We conducted comparisons with **DAC** in Table 7. Unfortunately, after reviewing **Selection-p** and **LanguaShrink**, we could not find open-source implementations for these methods. Since our main experiments use **GPT-4o** as the generative model, we cannot reliably reproduce or verify their performance under our main setup, including the task-agnostic scenario. However, to address the reviewer’s concern, we provide additional **task-aware evaluations** using **GPT-3.5-turbo**, where our method outperforms existing baselines. For LanguaShrink, we report the results as provided in their original paper.
> > >
> > > **GPT-3.5-turbo, 2k max context length, task-aware evaluation:**
> > >
> > > | Model        | SingleDoc | MultiDoc | Summarization | FewShot | Synth  | Code   | Avg     |
> > > | ------------ | --------- | -------- | ------------- | ------- | ------ | ------ | ------- |
> > > | LanguaShrink | 42.10      | 54.30     | 26.30          | 62.30    | 33.00   | 58.40   | 46.10    |
> > > | TPC-B        | 42.54   | 49.56  | 24.80       | 67.79 | 55.63 | 55.96  | 49.38 |
> > > | TPC-L        | 42.92     | 47.95    | 24.88         | 68.17   | 56.13 | 57.70 | 49.62 |
> > > | TPC-H        | 43.92     | 50.28 | 25.43       | 68.87   | 55.63 | 59.53  | 50.61 |
> > >
> > > These results show that, in the **task-aware setup**, our method outperforms current SOTA prompt compression methods (CPC: AVG=49.5, LanguaShrink: AVG=46.1, LLMLingua-2: AVG=39.1).
> > >
> > > We also provide **task-agnostic evaluations** for our methods and for competitors applicable to this setting:
> > >
> > > | Base Model  | SingleDoc | MultiDoc | Summarization | FewShot | Synth  | Code   | Avg     |
> > > | ----------- | --------- | -------- | ------------- | ------- | ------ | ------ | ------- |
> > > | LLMLingua-1 | 19.02   | 26.84  | 19.55       | 50.72   | 6.25   | 38.99 | 26.90 |
> > > | LLMLingua-2 | 18.38     | 6.86  | 22.09       | 20.78   | 0.78   | 31.11 | 16.67 |
> > > | TPC-B       | 40.67   | 47.09    | 23.44       | 60.98 | 31.66 | 39.64 | 40.58 |
> > > | TPC-L       | 37.65   | 42.99  | 23.74       | 59.36   | 30.75  | 37.38 | 38.64 |
> > > | TPC-H       | 42.93   | 47.47  | 23.30       | 61.47 | 43.38 | 40.93  | 43.25 |
> > >
> > > These evaluations confirm that our method maintains strong performance in both **task-aware** and **task-agnostic** setups, outperforming baselines where applicable.
> > >
> > >
> > > Also we provide the results for Llama3.1-8B-Instruct for the context length 2k for both setups.
> > > Task aware setup:
> > > | base_model                                     |   singledoc |   multidoc |   summarization |   fewshot |   synth |   code |     avg |
> > > |:-----------------------------------------------|------------:|-----------:|----------------:|----------:|--------:|-------:|--------:|
> > > | LLMLingua-1                                    |       13.86 |    17.09 |         20.42 |   51.74   |    4.94|  38.73 | 24.46 |
> > > | LLMLingua-2                                    |       19.17 |    24.65 |         21.31 |   54.39 |   12.93|  33.96| 27.73 |
> > > | TPC-B                                          |       36.56 |    42.48 |         25.05 |   65.29 |   48.85 |  41.81| 43.34 |
> > > | TPC-L                                          |       35.26 |    39.91 |         24.89 |   65.15 |   49.13|  41.85| 42.70 |
> > > | TPC-H                                          |       36.68 |    43.37 |         25.38 |   66.38   |   47.28 |  42.07| 43.52 |
> > >
> > > Task agnostic setup:
> > > | base_model                                      |   singledoc |   multidoc |   summarization |   fewshot |   synth |   code |     avg |
> > > |:------------------------------------------------|------------:|-----------:|----------------:|----------:|--------:|-------:|--------:|
> > > | LLMLingua-1                                     |     10.95   |     9.68   |         20.30 |   48.56 |    2.33|  34.67 | 21.08 |
> > > | LLMLingua-2                                     |     15.85 |     4.37|         17.02 |   19.58 |    1.68|  23.85| 13.72 |
> > > | TPC-B                                           |     33.40    |    33.41 |         23.88   |   41.46 |   17.02|  32.60| 30.29 |
> > > | TPC-L                                           |     31.93 |    33.42 |         22.92 |   25.94   |   22.79 |  31.02| 28.00 |
> > > | TPC-H                                           |     26.11 |    17.49 |         20.52   |   45.24   |   33.72 |  34.72 | 29.63 |
> > >
> > >
> > > All these new experiments are now added to page 21 of the revised paper.

---

> > > > ### Author Response · Authors · 2025-11-02
> > > >
> > > > > Expand evaluations in Tables 1-2 to include additional LLMs (e.g., GPT-3.5-Turbo, Llama-3-8B).
> > > >
> > > >
> > > > We provide additional evaluations for GPT-3.5-turbo and Llama3.1-8B-Instruct on LongBench dataset. The results are presented below. They are also now added to pages 21 of the paper.
> > > >
> > > >
> > > > **GPT-3.5-turbo, 2k max context length, task-aware evaluation:**
> > > >
> > > > | Model        | SingleDoc | MultiDoc | Summarization | FewShot | Synth  | Code   | Avg     |
> > > > | ------------ | --------- | -------- | ------------- | ------- | ------ | ------ | ------- |
> > > > | LanguaShrink | 42.10      | 54.30     | 26.30          | 62.30    | 33.00   | 58.40   | 46.10    |
> > > > | TPC-B        | 42.54   | 49.56  | 24.80       | 67.79 | 55.63 | 55.96  | 49.38 |
> > > > | TPC-L        | 42.92     | 47.95    | 24.88         | 68.17   | 56.13 | 57.70 | 49.62 |
> > > > | TPC-H        | 43.92     | 50.28 | 25.43       | 68.87   | 55.63 | 59.53  | 50.61 |
> > > >
> > > >
> > > > **GPT-3.5-turbo, 2k max context length, task-agnostic evaluation:**
> > > >
> > > > | Model  | SingleDoc | MultiDoc | Summarization | FewShot | Synth  | Code   | Avg     |
> > > > | ----------- | --------- | -------- | ------------- | ------- | ------ | ------ | ------- |
> > > > | LLMLingua-1 | 19.02   | 26.84  | 19.55       | 50.72   | 6.25   | 38.99 | 26.90 |
> > > > | LLMLingua-2 | 18.38     | 6.86  | 22.09       | 20.78   | 0.78   | 31.11 | 16.67 |
> > > > | TPC-B       | 40.67   | 47.09    | 23.44       | 60.98 | 31.66 | 39.64 | 40.58 |
> > > > | TPC-L       | 37.65   | 42.99  | 23.74       | 59.36   | 30.75  | 37.38 | 38.64 |
> > > > | TPC-H       | 42.93   | 47.47  | 23.30       | 61.47 | 43.38 | 40.93  | 43.25 |
> > > >
> > > >
> > > > **Llama3.1-8B-Instruct, 2k max context length, task-aware evaluation:**
> > > > | base_model|   singledoc |   multidoc |   summarization |   fewshot |   synth |   code |     avg |
> > > > |-|-|-|-|-|-|-|-|
> > > > | LLMLingua-1|       13.86 |    17.09 |         20.42 |   51.74   |    4.94|  38.73 | 24.46 |
> > > > | LLMLingua-2|       19.17 |    24.65 |         21.31 |   54.39 |   12.93|  33.96| 27.73 |
> > > > | TPC-B|       36.56 |    42.48 |         25.05 |   65.29 |   48.85 |  41.81| 43.34 |
> > > > | TPC-L|       35.26 |    39.91 |         24.89 |   65.15 |   49.13|  41.85| 42.70 |
> > > > | TPC-H|       36.68 |    43.37 |         25.38 |   66.38   |   47.28 |  42.07| 43.52 |
> > > >
> > > >
> > > > **Llama3.1-8B-Instruct, 2k max context length, task-agnostic evaluation:**
> > > > | base_model|   singledoc |   multidoc |   summarization |   fewshot |   synth |   code |     avg |
> > > > |-|-|-|-|-|-|-|-|
> > > > | LLMLingua-1|     10.95   |     9.68   |         20.30 |   48.56 |    2.33|  34.67 | 21.08 |
> > > > | LLMLingua-2|     15.85 |     4.37|         17.02 |   19.58 |    1.68|  23.85| 13.72 |
> > > > | TPC-B|     33.40    |    33.41 |         23.88   |   41.46 |   17.02|  32.60| 30.29 |
> > > > | TPC-L|     31.93 |    33.42 |         22.92 |   25.94   |   22.79 |  31.02| 28.00 |
> > > > | TPC-H|     26.11 |    17.49 |         20.52   |   45.24   |   33.72 |  34.72 | 29.63 |
> > > >
> > > >
> > > > > In Tables 1-2 and Figure 1, report uncompressed baselines
> > > >
> > > > We provide the results for GPT-4o and Llama-3.1-8B-Instruct on LongBench with uncompressed prompts.
> > > > We do not provide the results of GPT-3.5-turbo because that model cannot be applied to the **LongBench** benchmark due to its limited context window (4K tokens).
> > > >
> > > > We observe that our method demonstrates strong **prompt compression efficiency**. In particular, for the **Llama-3.1-8B-Instruct** model, performance degradation in the **task-agnostic** setting after compression is significantly smaller compared to other methods. Specifically, **TPC-Base (0.5B parameters)** achieves an overall score of **30.29** on LongBench, compared to **36.01** obtained by LLaMA with the full context.
> > > >
> > > > Moreover, in the **task-aware** setting, prompt compression with our method even leads to a **performance improvement**, highlighting the effectiveness of our **CSE module**.
> > > >
> > > > | base_model|   singledoc |   multidoc |   summarization |   fewshot |   synth |   code |     avg |
> > > > |-|-|-|-|-|-|-|-|
> > > > | GPT-4o |       45.32 |    65.90 |         26.94 |   70.47 |      60.00 | 73.99 | 57.10 |
> > > > | Llama-3.1-8B-Instruct |       44.83 |    46.10 |         24.14 |     34.48 |   53.26 | 13.27 | 36.01 |
> > > >
> > > > The revised manuscript now includes these results: GPT-4o in Table 1 (page 7) and Llama‑3.1‑8B‑Instruct in Table 9 (page 21).
> > > >
> > > > > Briefly explain why TPC-Base outperform TPC-Large on most tasks.
> > > >
> > > > We trained **TPC-Base** using **Qwen-2.5-0.5B** and **TPC-Large** using **LLaMA-3-1B** to demonstrate the scalability of our method across different LLM families. Since these models belong to different architectures—and Qwen is known for strong contextual understanding—we suspect that TPC-Base benefits from a higher degree of context comprehension, which explains why it outperforms TPC-Large on most tasks despite being smaller. The discussion is now added to page 11 of the paper.

---

### Review · Reviewer_xXkJ · 2025-10-10

**Summary Of Contributions:**

The authors claim the paper proposes a novel reward function, reinforcement based fine tuning, data curation for two datasets, and a task-agnostic prompt compression method.

In my opinion, the contribution that can be claimed in this paper is only the generation of task description from a long prompt to replace explicit questions.

- The reward function is not novel, it is KL divergence.
- The fine-tuning is not reinforcement learning.
- The datasets are not curated but generated by an LLM.
- The compression pipeline is described in another previous work.

**Audience:**

Yes

**Audience Explanation:**

Although not all the contributions proposed by the authors can be claimed as part of novel work, the question generation improves on the CPC results. This could be considered as an iterative advancement of an existing work. In other words, an extension of CPC.

**Broader Impact Concerns:**

If the utilized pre-trained LLM models already have biases, using them to generate synthetic data without human involvement and using them to train another model risks amplifying those biases.

Although prompt compression aims to reduce inference time and cost, the total pipeline which includes training multiple LLMs to compress the original prompt might increase the overall footprint.

**Claims And Evidence:**

No

**Claims Explanation:**

The authors claim that the work is introducing RL based fine-tuning that uses a novel reward function and cross-entropy loss.
I would not call the methodology described in the paper a RL. It is not clear how this method could be considered a RL approach (other than using the phrase "reward function"). The methodology describes a cross-entropy loss and the provided equations are not cross entropy. The novel reward function the authors presented is KL divergence. The utilization of KL in this task might be novel but the function cannot be claimed as novel.

Furthermore, in order to claim data curation as contribution, I would expect to see some of the claims made in the paper to be analyzed and presented. The data generated are provided by pre-trained LLMs. Authors claim that the pipeline proposed generates diverse pairs for q-p, the questions require reasoning and the positives do not contain entire information required to answer the question. The data generated were not analyzed against these claims. The paper does not contain proof of such claims.

Moreover, the paper suggest that the pipeline is novel, especially in providing a task agnostic pipeline. The only difference between the SOTA method (CPC) is that the pipeline utilizes an LLM to generate the questions instead of the explicit questions given in the input. The rest of the pipeline is the same as CPC. I do not see the contribution in the general pipeline.

**Requested Changes:**

I believe that the following are crucial adjustments or clarifications needed for this paper:

The reward function is not novel, it is KL divergence.
- The authors need to clarify why they claim KL divergence is a new method.

The fine-tuning is not reinforcement learning.
- The authors need to clarify how their approach could be considered an RL methodology

The datasets are not curated but generated by an LLM.
- The authors need to provide extended analysis of the data to provide evidence that the generated synthetic data is balanced, provides diverse examples/pairs, requires reasoning with context, and the positives do not contain entire information required to answer the question.

The compression pipeline is described in another previous work.
- The authors need to provide enough evidence that this work is significantly different than CPC while highlighting the novel contributions beyond CPC.


Furthermore, I noticed that especially the related work section highly resembles the related work section of CPC paper. The authors need to address this.

---

> ### Author Response · Authors · 2025-11-02
>
> We would like to thank the reviewer for their thoughtful feedback and comments. Below we have answered the questions and reflected the changes in the revised manuscript in *blue*. Please don't hesitate to let us know if anything was unclear or you required additional clarification.
>
>
> > The reward function is not novel, it is KL divergence.
>
>
> Although the KL divergence itself is not novel, our reward formulation of using the KL divergence as the reward function for prompt compression is novel. Specifically, we first generate a set of **task descriptions** for a given prompt *p*. For each description, we then produce a compressed context using the CSE module. Knowing the ideal reference answer, we compute the KL divergence between the conditional probability distributions of generating that answer given the **compressed** and **original** prompts.
>
> Importantly, the reward is applied **not** to the compressed prompts or their generated answers directly, but rather to the **task descriptions** that guided the compression process. This design allows the learning signal to propagate to an earlier stage of the compression pipeline, effectively improving the overall policy.
>
> We chose KL divergence due to its generality and robustness across different domains, making it a suitable and scalable metric for our setup.
>
> > The fine-tuning is not reinforcement learning
>
>
> We employ the concept of **Cross-Entropy Reinforcement Learning (CERL) (Rubinstein, 1999 [1])**, which is considered an RL method, where the agent generates multiple actions at each step, and the top percentile of the best-performing actions—according to the reward—is selected. The agent is then fine-tuned to reproduce these actions in the given state.
>
> In our setup, an action corresponds to generating a **task description**. Over multiple iterations, we maintain a pool of prompts, generate several task descriptions for each, select the best-performing one, and train the CTD module on the resulting **prompt–best-task-description pairs**. Across epochs, the task descriptions gradually become more effective for context compression, demonstrating the efficacy of our method.
>
> Citations:
>
> [1] The Cross‑Entropy Method for Combinatorial and Continuous Optimization, by Reuven Y. Rubinstein (1999) — Methodology and Computing in Applied Probability, Vol. 1(2), pp. 127-190. DOI: 10.1023/A:1010091220143.

---

> > ### Author Response · Authors · 2025-11-02
> >
> > > The datasets are not curated but generated by an LLM.
> >
> > > The authors need to provide extended analysis of the data to provide evidence that the generated synthetic data is balanced, provides diverse examples/pairs, requires reasoning with context, and the positives do not contain the entire information required to answer the question.
> >
> > Although we use an LLM to generate prompts for the dataset, we perform human verification to ensure the absence of errors and artifacts. We recognize that synthetic data generation can introduce biases or limit diversity. To mitigate this, our datasets were curated from multiple sources, adapted to instructions, and subjected to **multi-stage prompting** to improve coverage and structure. Furthermore, our **reinforcement learning refinement stage** helps align task descriptions with the resulting outputs, serving as a practical safeguard against overfitting to synthetic artifacts.
> >
> > To provide a more convincing assessment of generalization, we include additional **OOD evaluations**, which demonstrate the strong generalization capability of our method. We also use an **LLM-as-a-Judge evaluation technique** to compare our approach against other methods.
> >
> > Specifically, we evaluate the compressed prompts along the following dimensions:
> > * Coherency: We ask an LLM to rate coherence on a scale from 1 to 5.
> > * Intent preservation: We evaluate how well the compressed prompt retains the user's original intent.
> > * Contextual information preservation: We assess the extent to which the compressed prompt enables the LLM to generate a response similar to what it would produce when given the original prompt.
> >
> > The evaluations are conducted on the following out-of-distribution datasets:
> >
> > * Financial News Prompts(FNP), a set of complex financial prompts.
> > * Self-Instruct, which includes complex prompts generated by LLM
> > * P3, a subset of crowd-sourced instructions and demonstrations sampled from the Public Pool of Prompts (P3) dataset
> > * UltraChat, a subset of Long instruction-following prompts curated for UltraLM. Examples include multi-paragraph instructions, simulated conversations, and educational tasks.
> > * SystemCheck, a subset of the dataset containing generated user queries targeting 14 test-only system prompts with irrelevant in-context task demonstrations.
> > * A subset of OpenOrca, extended OpenAI prompts (via ShareGPT + FLAN), many with long, deeply nested reasoning tasks.
> >
> > Because the original datasets are quite large (e.g., OpenOrca ≈3M examples), we sample 200 examples per dataset uniformly at random for experiments.
> >
> > As shown in the table below, our method consistently achieves higher scores than existing approaches. Additionally, for each original instruction prompt from a given dataset, we directly ask the GPT-4.1 LLM to choose between the compressed prompt generated by our method and that produced by a competing method. We define the win rate of compressed prompt A over B as the number of times A is preferred over B, divided by the total number of samples from that dataset. A higher win rate thus reflects a more effective prompt compression strategy. According to the evaluation results, GPT-4.1 consistently prefers prompts compressed by our method over those produced by LLMLingua-2. We also observe a substantial performance advantage over other task-agnostic prompt compression baselines. We have now added these new experiments to pages 20-22 (section A.5) of the revised paper.
> >
> > TPC vs LLMLingua-2:
> > |Dataset|Method|Coherency|Intent|Context|Winrate|
> > |-|-|-|-|-|-|
> > |FNP|LLMLingua-2|2.5|2.5|1.6|0.6|
> > |FNP|TPC|3.6|4.7|3.4|0.4|
> > |Self-Instruct|LLMLingua-2|2.2|2.5|1.8|0.2|
> > |Self-Instruct|TPC|3.6|2.8|2.4|0.8|
> > |P3|LLMLingua-2|3.2|3.6|2.9|0.4|
> > |P3|TPC|2.9|2.8|2.3|0.6|
> > |UltraChat|LLMLingua-2|2.4|2.9|2.3|0.1|
> > |UltraChat|TPC|3.7|3.4|2.7|0.9|
> > |SystemCheck|LLMLingua-2|1.8|2.8|2.4|0.1|
> > |SystemCheck|TPC|3.6|3.1|2.3|0.9|
> > |OpenOrca|LLMLingua-2|2.8|3.1|2.5|0.1|
> > |OpenOrca|TPC|3.4|3.3|2.4|0.9|
> >
> > TPC vs SelectiveContext:
> > |Dataset|Method|Coherency|Intent|Context|Winrate|
> > |-|-|-|-|-|-|
> > |FNP|SelectiveContext|2.6|2.6|1.8|0.1|
> > |FNP|TPC|3.6|4.7|3.4|0.9|
> > |Self-Instruct|SelectiveContext|2.7|3.1|2.1|0.1|
> > |Self-Instruct|TPC|3.6|2.8|2.4|0.9|
> > |P3|SelectiveContext|3.2|3.1|2.7|0.1|
> > |P3|TPC|2.9|2.8|2.3|0.9|
> > |UltraChat|SelectiveContext|2.3|2.4|1.7|0.0|
> > |UltraChat|TPC|3.7|3.4|2.7|1.0|
> > |SystemCheck|SelectiveContext|1.9|2.4|1.6|0.0|
> > |SystemCheck|TPC|3.6|3.1|2.3|1.0|
> > |OpenOrca|SelectiveContext|2.7|3.0|2.3|0.2|
> > |OpenOrca|TPC|3.4|3.3|2.4|0.8|
> >
> > TPC vs LLMLingua-1:
> > |Dataset|Method|Coher.|Intent|Context|Winrate|
> > |-|-|-|-|-|-|
> > |FNP|LLMLingua-1|1.7|1.6|1.2|0.0|
> > |FNP|TPC|3.6|4.7|3.4|1.0|
> > |Self-Instruct|LLMLingua-1|2.2|2.4|2.0|0.0|
> > |Self-Instruct|TPC|3.6|2.8|2.4|1.0|
> > |P3|LLMLingua-1|1.9|2.6|1.8|0.0|
> > |P3|TPC|2.9|2.8|2.3|1.0|
> > |UltraChat|LLMLingua-1|1.5|1.7|1.3|0.0|
> > |UltraChat|TPC|3.7|3.4|2.7|1.0|
> > |SystemCheck|LLMLingua-1|1.5|1.6|1.5|0.0|
> > |SystemCheck|TPC|3.6|3.1|2.3|1.0|
> > |OpenOrca|LLMLingua-1|1.8|2.2|1.7|0.0|
> > |OpenOrca|TPC|3.4|3.3|2.4|1.0|

---

> > > ### Author Response · Authors · 2025-11-02
> > >
> > > > The compression pipeline is described in another previous work.
> > >
> > > > The authors need to provide enough evidence that this work is significantly different than CPC while highlighting the novel contributions beyond CPC.
> > >
> > > Our compression algorithm is significantly different/novel from CPC, as it is a task-agnostic prompt compression method that consists of the following novel ideas:
> > >
> > > 1. **A novel module for assessing the relevance of text segments to arbitrary task descriptions.** Unlike CPC, which could only determine relevance for semantically coherent sentences and heavily depended on a sentence-splitting strategy, our module can operate on texts of arbitrary structure. Moreover, we demonstrate that comparable performance can be achieved with a model **14× smaller** (0.5B for TPC-Base vs 7B parameters for CPC).
> > >
> > > 2. **A new task formulation: task-agnostic prompt compression.** This setup requires the compression method to work universally, without additional manual intervention from the user.
> > >
> > > 3. We introduce the **CTD component**, which guides prompt compression by generating effective and concise task descriptions. Notably, this module requires almost no additional GPU memory compared to **CSE**, as it shares the same underlying language model and adds only a small **LoRA** layer on top.
> > >
> > > 4. **A method of synergistic training of two modules.** This approach allows compression of prompts of any structure. Additionally, we effectively leverage **reinforcement learning** to train the module that generates arbitrary task descriptions, enabling strong **OOD scalability**.
> > >
> > > We have now added a summary of this discussion at the end of the Introduction section on page 2 of the revised paper.

---

### Review · Reviewer_eS4v · 2025-10-12

**Summary Of Contributions:**

The paper deals with prompt compression, the task of reducing prompt length while retaining the important information to provide accurate answers, thus reducing inference time and costs. Unlike various recent approaches that require explicit questions, the paper focuses on the task-agnostic setting. Specifically, the proposed approach trains a module that generates context-relevant task descriptions in lieu of the explicit questions and fine-tune this module using RL with reward function that aims to improve the final generated answer. Experiments on LongBench and ZeroSCROLLS show gains over the baselines, particularly in the prompt-agnostic setting.

Strengths:
- The problem addressed is important and of interest
- Overall, the paper is written well and the technical approach is clearly described
- The experiments show improvement over the baselines, in both task-aware and task-agnostic (with gains that are much more significant in the task-agnostic setting).


Weaknesses:
1. The importance of the context-agnostic setting could be further explained/motivated: the paper notes that approaches like CPC "cannot be adapted to tasks with no explicit questions, such as summarization”. This claim could be explained in more details: why is there no question in summarization tasks? There should be some instruction that indicates that the goal is to summarize the text. Why can't this instruction be used as the question in the CPC pipeline?
2. Not clear why RL is applied specifically to CTD. Given that the RL training is operating on the complete task, taking into account the result of the CSE module, it is not clear why the RL training is limited to operating on the CTD module. For example, the same RL training could have been operating on the CSE module (leaving the CTD module frozen) or on both.
3. Simple baselines for CTD is missing, namely simply using the result of a strong LLM (e.g., GPT-4o, Claude 4 Sonnet, DeepSeek V3/R1) prompted to extract a question (which seems similar to how the CTD training data was created, although not clear what model was used for that purpose). While in this case we will not be able to apply the RL training on the LLM-based CTD component, as noted in my previous comment, it sounds like it may be equally applicable on the CSE module.
4. Similarly, LLM-based task description together with CPC would have been a very useful baseline for the task-agnostic setting.
5. Not clear how to interpret the ablation analyses: why the max result in Table 3(a) and Table 3(b) are different? The top row in both Table should be representing the full setting of CTD and CSE respectively, assuming the other component is also provided in full setting. So it is not clear to me why the performance in the full setting is not the same for both tables.
6. The results presented for some baselines do not match their original papers.Specifically, there are very large gaps between the results reported for LLMLingua-2 in the manuscript vs. the original paper: SingleDoc (35.5 vs 20.18), MultiDoc (38.7 vs. 7.0), Few-Shot (69.6 vs. 13.02), etc. This is very concerning and I could not find any explanation for these gaps in the paper.

**Audience:**

Yes

**Audience Explanation:**

As noted in the strength above, the topic is important and likely to be of interest to various individuals in TMLR's audience as: (1) it has the potential of reducing inference time and costs in LLMs; (2) it builds on and compare to published literature in the field.

**Broader Impact Concerns:**

I have no concerns.

**Claims And Evidence:**

No

**Claims Explanation:**

As per the list of weaknesses above, I have concerns about insufficient evidence provided in the paper to support the presented approach and the claims about its performance, specifically w.r.t the importance/relevance of this setting (weakness #1), the need/usefulness for RL in the proposed setting (weakness #2), the importantly the experimental evaluation including the selected baselines, the reported results for the baselines, and the ablation study (Weaknesses #3, #4, #5, #6).

**Requested Changes:**

Based on my comments above, the requested changes are:
- Clarify the motivation for the proposed task-agnostic setting
- Explain the gaps in performance of baselines
- Describe the role of RL and whether it could have been applied to the CSE module or both
- Add simpler baselines for CTD and for the task-agnostic CPC, or explain why they are not applicable
- Clarify the differences between the full setting in the ablation study of both CTD and CSE and why they are not conducted starting from the same full setting

---

> ### Author Response · Authors · 2025-11-02
>
> We would like to thank the reviewer for their thoughtful feedback and comments. Below we have answered the questions and reflected the changes in the revised manuscript in *blue*. Please don't hesitate to let us know if anything was unclear or you required additional clarification.
>
>
> > The importance of the context-agnostic setting could be further explained/motivated.
>
> Methods such as CPC expect both the text and an explicit question as input, using the question to assess the relevance of different text segments. This becomes a limitation, since in many real-world scenarios, users of prompt compression methods may not be able or may not wish to formulate an appropriate question to accompany the text. In contrast, our method enables context compression without additional manual input or task-specific configurations. In our work, we define a task-agnostic setup, where the prompt compression method receives a text input without any explicitly marked components that can or cannot be compressed. The compression model must automatically determine which parts of the prompt are important and to what extent they should be compressed. In this setup, the model must infer the domain and structure of the prompt autonomously. We have now added this motivating discussion to pages 3-4 of the revised paper.
>
> > Why is RL applied specifically to CTD, and not CSE?
>
> Since the CSE module is inherently non-generative (it assigns a numerical score to each sentence in the context, estimating its relevance to the query) applying RL to it would effectively reduce to learning to maximize the selection of relevant segments while minimizing the inclusion of irrelevant ones. This process is conceptually equivalent to the contrastive training already performed during the pretraining of the CSE module. In that stage, the CSE learns to distinguish relevant context segments from irrelevant ones by producing higher scores for the former and lower scores for the latter. Therefore, applying RL to the CSE module would largely duplicate the learning dynamics of its pretraining stage.
>
> > LLM-based task description together with CPC would have been a very useful baseline
>
> To address this comment, we include additional experiments using an LLM to synthetically generate short questions, which are then provided to the CPC. All experiments were conducted on the **LongBench** benchmark using **GPT-4o** as the generation model. The results are presented below.
>
> | Setting           | SingleDoc | MultiDoc | Summarization | FewShot | Synth  | Code   |
> | ----------------- | --------- | -------- | ------------- | ------- | ------ | ------ |
> | **Task-aware**    | 40.05     | 56.31    | 25.30         | 27.42   | 16.59 | 29.14  |
> | **Task-agnostic** | 4.60     | 1.25     | 11.55        | 0.00     | 0.00    | 19.51 |
>
>
> As we can see, our simple use of an LLM to generate task descriptions combined with CPC for selecting relevant chunks of the prompt performs reasonably on the task-aware setup, but on the task-agnostic setup it shows very poor results, indicating the low stability of this approach to prompt distortion during compression. We have now added this experiment to page 13 (Table 5) of the revised paper.

---

> > ### Author Response · Authors · 2025-11-02
> >
> > > The results presented for some baselines do not match their original papers
> >
> > The reason for the discrepancy comes from the change in the pre-trained LLM. LLMLingua* evaluated their models using **GPT-3.5-turbo**, which is considerably outdated. Therefore, we evaluated the existing methods and our proposed method on the latest **GPT-4o** model. To further address the reviewer’s concern, we also report the results obtained with **GPT-3.5-turbo**, shown below for our method as well. Additionally, we conduct experiments under two setups: **task-aware prompt compression** and **task-agnostic prompt compression**, while all existing methods reported only the results for the first setup.
> >
> > In the **task-aware** setup, the model compresses only a selected part of the context, which is then embedded into a predefined template consisting of a system prompt, a question, and other structural elements before being passed to the generative model for evaluation. For example:
> >
> > ```
> > You are a smart assistant. You will be given several texts which are in mixed order. Provide the correct order of the texts’ ordering.
> >
> > Text 1: [[context to compress…]]
> > Text 2: [[context to compress…]]
> > …
> > Text N: [[context to compress…]]
> >
> > Your response:
> > ```
> >
> > Here, only the text inside the brackets is subject to compression.
> >
> > In the **task-agnostic** setup, the model is asked to compress the *entire* prompt, including the instructions and the context, so that it can be directly passed to the LLM without any additional templating or manual modifications:
> >
> > ```
> > [[You are a smart assistant. You will be given several texts which are in mixed order. Provide the correct order of the texts.
> >
> > Text 1: context to compress…
> > Text 2: context to compress…
> > …
> > Text N: context to compress…
> >
> > Your response:]]
> > ```
> >
> > In their experiments, LLMLingua-* used the first (task-aware) setup. However, we argue that this does not represent a truly fair prompt compression scenario since it relies on external manual structuring. For completeness, we report the results for both setups in our paper.
> >
> > **Task-aware setup (context length 2k, GPT-3.5-turbo):**
> >
> > | Model | SingleDoc | MultiDoc | Summarization | FewShot | Synth |  Code |   Avg |
> > | :---- | --------: | -------: | ------------: | ------: | ----: | ----: | ----: |
> > | TPC-L |     42.92 |    47.95 |         24.88 |   68.17 | 56.13 | 57.70 | 49.62 |
> > | TPC-B |     42.54 |    49.56 |         24.80 |   67.79 | 55.63 | 55.96 | 49.38 |
> > | TPC-H |     43.92 |    50.28 |         25.43 |   68.87 | 55.63 | 59.53 | 50.61 |
> >
> > **Task-agnostic setup (context length 2k, GPT-3.5-turbo):**
> >
> > | Model       | SingleDoc | MultiDoc | Summarization | FewShot | Synth |  Code |   Avg |
> > | :---------- | --------: | -------: | ------------: | ------: | ----: | ----: | ----: |
> > | LLMLingua-1 |     19.02 |    26.84 |         19.55 |   50.72 |  6.25 | 38.99 | 26.90 |
> > | LLMLingua-2 |     18.38 |     6.86 |         22.09 |   20.78 |  0.78 | 31.11 | 16.67 |
> > | TPC-L       |     37.65 |    42.99 |         23.74 |   59.36 | 30.75 | 37.38 | 38.64 |
> > | TPC-B       |     40.67 |    47.09 |         23.44 |   60.98 | 31.66 | 39.64 | 40.58 |
> > | TPC-H       |     42.93 |    47.47 |         23.30 |   61.47 | 43.38 | 40.93 | 43.25 |
> >
> > These results confirm that our reported scores are consistent when evaluated under comparable conditions and that the observed discrepancies primarily stem from differences in the setup and model version, rather than methodological inconsistencies. These experiments are now added to page 21 (Table 8) of the revised paper.

---

### Comment · Action_Editor_gS9c · 2025-10-12
**Start author-reviewer discussions**

Dear authors,

You have three reviews. Please address their comments as soon as possible. The discussion period will end on October 26th, 2025.

I'd suggest that you share author rebuttals when ready. October 26th, 2025 is the end of the **discussion** period. If you will want to reply to reviewers' response to your rebuttal, it would be better to share your author rebuttals as soon as possible.

---

Dear reviewers,

Thank you for submitting your reviews. As this paper received three reviews, now it's time to have discussions with authors.

You will be able to submit your formal decision recommendation starting in 2 weeks. Your prompt responses to authors' comments would be very appreciated.

Remember that different from other journals / conferences, [TMLR's acceptance criteria](https://jmlr.org/tmlr/reviewer-guide.html) are based on positive answers to the following two questions:
- Are the claims made in the submission supported by accurate, convincing and clear evidence?
- Would some individuals in TMLR's audience be interested in the findings of this paper?

Novelty of the studied method is not a necessary criterion for acceptance at TMLR.

It's still not too late to review [the TMLR's reviewer guideline](https://jmlr.org/tmlr/reviewer-guide.html) in case you are not aware of that.


Thank you!
Your AE

---

### Comment · Action_Editor_gS9c · 2025-11-03
**Submit your official recommendation**

Dear Reviewers,

Now it's time for you to submit an official recommendation for this submission.

The authors responded to your initial reviews.
Please
1. consider their rebuttal comments,
2. respond to the comments ***ASAP*** if you have follow-up questions or comments, and
3. submit your official recommendation ***by November 16th***, including ***your thoughts on the author rebuttal***, specifically, what point in your review their responses (did not) address. It will greatly help me write a meta-review and make a fair decision.

Thank you,
Your AE

---

### Comment · Action_Editor_gS9c · 2025-12-04
**Changes in camera-ready**

Dear authors,

Could you please clarify which part of references section you updated?

I also find the following preprint is published at IEEE Access
- Languashrink: Reducing token overhead with psycholinguistics. Xuechen Liang, Meiling Tao, Yinghui Xia, Tianyu Shi, Jun Wang, JingSong Yang (arXiv).
- LinguaShrink: Reducing Token Overhead With Psycholinguistics. Feihu Ma; Xuechen Liang; Meiling Tao (IEEE Access)

, which looks weird since the author list does not exactly match the preprint's one

Thank you,
Your AE

---

> ### Author Response · Authors · 2025-12-05
>
> We'd like to thank AE for his thoughtful comment. Below is a list of references we've updated to point to official publications:
>
> | arXiv reference| venue reference|
> |-|-|
> | Bai, Yushi; Lv, Xin; Zhang, Jiajie; Lyu, Hongchang; Tang, Jiankai; Huang, Zhidian; Du, Zhengxiao; Liu, Xiao; Zeng, Aohan; Hou, Lei; Dong, Yuxiao; Tang, Jie; Li, Juanzi. “Longbench: A bilingual, multitask benchmark for long context understanding.” arXiv:2308.14508 (2023). | Bai, Yushi; Lv, Xin; Zhang, Jiajie; Lyu, Hongchang; Tang, Jiankai; Huang, Zhidian; Du, Zhengxiao; Liu, Xiao; Zeng, Aohan; Hou, Lei; Dong, Yuxiao; Tang, Jie; Li, Juanzi. “LongBench: A Bilingual, Multitask Benchmark for Long Context Understanding.” Proceedings of the 62nd Annual Meeting of the Association for Computational Linguistics (ACL 2024), pp. 3119–3137. |
> | Dong, Qingxiu; Li, Lei; Dai, Damai; Zheng, Ce; Wu, Zhiyong; Chang, Baobao; Sun, Xu; Xu, Jingjing; Sui, Zhifang. “A Survey on In-context Learning.” arXiv:2301.00234 (2022).                                                                                                     | Dong, Qingxiu; Li, Lei; Dai, Damai; Zheng, Ce; Ma, Jingyuan; Li, Rui; Xia, Heming; Xu, Jingjing; Wu, Zhiyong; Chang, Baobao; Sun, Xu; Li, Lei; Sui, Zhifang. “A Survey on In-context Learning.” EMNLP 2024, pp. 1107–1128.|
> | Jiang, Huiqiang; Wu, Qianhui; Lin, Chin-Yew; Yang, Yuqing; Qiu, Lili. “LLMLingua: Compressing Prompts for Accelerated Inference of Large Language Models.” arXiv:2310.05736 (2023).| Jiang, Huiqiang; Wu, Qianhui; Lin, Chin-Yew; Yang, Yuqing; Qiu, Lili. “LLMLingua: Compressing Prompts for Accelerated Inference of Large Language Models.” EMNLP 2023, pp. 13358–13376. |
> | Pan, Zhuoshi; Wu, Qianhui; Jiang, Huiqiang; Xia, Menglin; Luo, Xufang; Zhang, Jue; Lin, Qingwei; Rühle, Victor; Yang, Yuqing; Lin, Chin-Yew; others. “LLMLingua-2: Data Distillation for Efficient and Faithful Task-Agnostic Prompt Compression.” arXiv:2403.12968 (2024).     | Pan, Zhuoshi; Wu, Qianhui; Jiang, Huiqiang; Xia, Menglin; Luo, Xufang; Zhang, Jue; Lin, Qingwei; Rühle, Victor; Yang, Yuqing; Lin, Chin-Yew; Zhao, H. Vicky; Qiu, Lili; Zhang, Dongmei. “LLMLingua-2: Data Distillation for Efficient and Faithful Task-Agnostic Prompt Compression.” Findings of ACL 2024, pp. 963–981.|
> | Wang, Cangqing; Yang, Yutian; Li, Ruisi; Sun, Dan; Cai, Ruicong; Zhang, Yuzhu; Fu, Chengqian; Floyd, Lillian. “Adapting LLMs for Efficient Context Processing Through Soft Prompt Compression.” arXiv:2404.04997 (2024).                                                        | Wang, Cangqing; Yang, Yutian; Li, Ruisi; Sun, Dan; Cai, Ruicong; Zhang, Yuzhu; Fu, Chengqian. “Adapting LLMs for Efficient Context Processing Through Soft Prompt Compression.” International Conference on Modeling, Natural Language Processing and Machine Learning 2024, pp. 91–97.|
> | BehnamGhader, Parishad; Adlakha, Vaibhav; Mosbach, Marius; Bahdanau, Dzmitry; Chapados, Nicolas; Reddy, Siva. “LLM2Vec: Large Language Models Are Secretly Powerful Text Encoders.” arXiv:2404.05961 (2024).                                                                    | BehnamGhader, Parishad; Adlakha, Vaibhav; Mosbach, Marius; Bahdanau, Dzmitry; Chapados, Nicolas; Reddy, Siva. “LLM2Vec: Large Language Models Are Secretly Powerful Text Encoders.” First Conference on Language Modeling (2024).|
> | Sanh, Victor; Webson, Albert; Raffel, Colin; Bach, Stephen H.; Sutawika, Lintang; Alyafeai, Zaid; Chaffin, Antoine; Stiegler, Arnaud; Scao, Teven; Raja, Arun; others. “Multitask prompted training enables zero-shot task generalization.” arXiv:2110.08207 (2021). | Sanh, Victor; Webson, Albert; Raffel, Colin; Bach, Stephen H.; Sutawika, Lintang; Alyafeai, Zaid; Chaffin, Antoine; Stiegler, Arnaud; Scao, Teven; Raja, Arun; others. “Multitask Prompted Training Enables Zero-Shot Task Generalization.” In *International Conference on Learning Representations (ICLR 2022)*.|
> | Ding, Ning; Chen, Yulin; Xu, Bokai; Qin, Yujia; Zheng, Zhi; Hu, Shengding; Liu, Zhiyuan; Sun, Maosong; Zhou, Bowen. “Enhancing chat language models by scaling high-quality instructional conversations.” arXiv:2305.14233 (2023).                                   | Ding, Ning; Chen, Yulin; Xu, Bokai; Qin, Yujia; Hu, Shengding; Liu, Zhiyuan; Sun, Maosong; Zhou, Bowen. “Enhancing Chat Language Models by Scaling High-quality Instructional Conversations.” In *Proceedings of the Conference on Empirical Methods in Natural Language Processing (EMNLP 2023)*.|
> | Holtzman, Ari; Buys, Jan; Du, Li; Forbes, Maxwell; Choi, Yejin. “The curious case of neural text degeneration.” arXiv:1904.09751 (2019).| Holtzman, Ari; Buys, Jan; Du, Li; Forbes, Maxwell; Choi, Yejin. “The Curious Case of Neural Text Degeneration.” In *Proceedings of the International Conference on Learning Representations (ICLR)*.|
>
> Due to comment length limits, we are publishing the second part of the list in the accompanying comment below.

---

> > ### Author Response · Authors · 2025-12-05
> >
> > Below we provide the second part of the list of updated references:
> > | arXiv reference                                                                                                                                                                                                                                                                     | venue reference                                                                                                                                                                                                                                                                                                                                                               |
> > |-|-|
> > | Zhao, Yi; Li, Zuchao; Zhao, Hai; Qi, Baoyuan; Liu, Guoming. “DAC: A Dynamic Attention-aware Approach for Task-Agnostic Prompt Compression.” arXiv:2507.11942 (2025).                                                                                                 | Zhao, Yi; Li, Zuchao; Zhao, Hai; Qi, Baoyuan; Liu, Guoming. “DAC: A Dynamic Attention-aware Approach for Task-Agnostic Prompt Compression.” In *Proceedings of the Annual Meeting of the Association for Computational Linguistics (ACL 2025)*, pages 19395–19407.                                                                                                                                                    |
> > | Wei, Jason; Bosma, Maarten; Zhao, Vincent Y.; Guu, Kelvin; Yu, Adams Wei; Lester, Brian; Du, Nan; Dai, Andrew M.; Le, Quoc V. “Finetuned language models are zero-shot learners.” arXiv:2109.01652 (2021).                                                           | Wei, Jason; Bosma, Maarten; Zhao, Vincent; Guu, Kelvin; Yu, Adams Wei; Lester, Brian; Du, Nan; Dai, Andrew M.; Le, Quoc V. “Finetuned Language Models are Zero-Shot Learners.” In *Proceedings of the International Conference on Learning Representations (ICLR)*.                                                                                                                                                   |
> > | Ivison, Hamish; Wang, Yizhong; Pyatkin, Valentina; Lambert, Nathan; Peters, Matthew; Dasigi, Pradeep; Jang, Joel; Wadden, David; Smith, Noah A.; Beltagy, Iz; others. “Camels in a changing climate: Enhancing lm adaptation with tulu 2.” arXiv:2311.10702 (2023). | Ivison, Hamish; Wang, Yizhong; Pyatkin, Valentina; Lambert, Nathan; Peters, Matthew E.; Dasigi, Pradeep; Jang, Joel; Wadden, David; Smith, Noah A.; Beltagy, Iz; others. “Camels in a Changing Climate: Enhancing LM Adaptation with Tulu 2.” CoRR (2023).                    |
> > | Xuechen Liang, Meiling Tao, Yinghui Xia, Tianyu Shi, Jun Wang, and JingSong Yang. Languashrink: Reducing token overhead with psycholinguistics. arXiv preprint arXiv:2409.00855, 2024 | Feihu Ma, Xuechen Liang, and Meiling Tao. Linguashrink: Reducing token overhead with psycholinguistics. IEEE Access, 12:145095–145105, 2024. doi: 10.1109/ACCESS.2024.3470512|
> >
> >
> > Also, in response to AE's suggestion, we reviewed both versions of the LanguaShrink paper and found no significant discrepancies in content. Although the author lists are not entirely identical, all available information on the overlapping set of authors is consistent, as is the content of both articles. Therefore, we have also replaced the corresponding arxiv publication with a link to the IEEE Access publication.

---

> > > ### Comment · Action_Editor_gS9c · 2025-12-05
> > >
> > > Dear Authors,
> > >
> > > Thank you for the clarifications. I just approved the camera-ready
> > > I also hope the code repository is populated soon as the repository is currently empty.
> > >
> > > Best,
> > > Your AE

---

### Decision · Action_Editor_gS9c · 2025-11-24

**Recommendation:** Accept with minor revision

**Additional Comments:**

Please review references section and replace arXiv papers with published papers. I found several papers cited as arXiv papers published at venues.

**Audience:**

Yes

**Audience Explanation:**

Task-agnostic prompt compression should be an interest of this community given that computation complexity to perform a task with LLMs has been increasing.

**Claims And Evidence:**

Yes

**Claims Explanation:**

Through the author rebuttal including additional experiments, three reviewers confirmed that their significant concerns were addressed and agreed that the empirical evidence supports the claims in this work. The rebuttal also cleared up some confusions in claims including fine-tuning vs. RL and missing baselines pointed out by the reviewers.